# Revocable Deep Reinforcement Learning with Affinity Regularization for Outlier-Robust Graph Matching

**Chang Liu**[1], **Zetian Jiang**[1], **Runzhong Wang**[1,2], **Lingxiao Huang**[3], **Pinyan Lu**[4,5], **Junchi Yan**[1,2*]

[1]MoE Key Lab of Artificial Intelligence, Shanghai Jiao Tong University     [2]Shanghai AI Laboratory

[3]State Key Laboratory of Novel Software Technology, Nanjing University

[4]Shanghai University of Finance and Economics     [5]Huawei TCS Lab

`{only-changer,maple_jzt,runzhong.wang,yanjunchi}@sjtu.edu.cn`
`huanglingxiao1990@126.com, lu.pinyan@mail.shufe.edu.cn`
PyTorch Code:   `https://github.com/Thinklab-SJTU/RGM.git`

## Abstract

Graph matching (GM) has been a building block in various areas including computer vision and pattern recognition. Despite recent impressive progress, existing deep GM methods often have obvious difficulty in handling outliers, which are ubiquitous in practice. We propose a deep reinforcement learning based approach RGM, whose sequential node matching scheme naturally fits the strategy for selective inlier matching against outliers. A revocable action framework is devised to improve the agent's flexibility against the complex constrained GM. Moreover, we propose a quadratic approximation technique to regularize the affinity score, in the presence of outliers. As such, the agent can finish inlier matching timely when the affinity score stops growing, for which otherwise an additional parameter i.e. the number of inliers is needed to avoid matching outliers. In this paper, we focus on learning the back-end solver under the most general form of GM: the Lawler's QAP, whose input is the affinity matrix. Especially, our approach can also boost existing GM methods that use such input. Experiments on multiple real-world datasets demonstrate its performance regarding both accuracy and robustness.

## 1 Introduction

Graph matching (GM) aims to find node correspondence between two or multiple graphs. As a standing and fundamental problem, GM spans wide applications in different areas including computer vision and pattern recognition. With the increasing computing resource, graph matching that involves the second-order edge affinity (in contrast to the linear assignment problem e.g. bipartite matching) becomes a powerful and relatively affordable tool for solving the correspondence problem with moderate size, and there is growing research in this area, especially with the introduction of deep learning in recent years (Zanfir et al., 2018; Wang et al., 2019b).

GM can be formulated as a combinatorial optimization problem namely Lawler's Quadratic Assignment Problem (Lawler's QAP) (Lawler, 1963), which is known as NP-hard. Generally speaking, handling the graph matching problem involves two steps: extracting features from input images to formulate a QAP instance and solving that QAP instance via constrained optimization, namely **front-end** feature extractor and **back-end** solver, respectively. Impressive progress has been made for graph matching with the introduction of rich deep learning techniques. However, in existing deep GM works, the deep learning modules are mainly applied on the front-end, especially for visual images using CNN for node feature learning (Zanfir et al., 2018) and GNN for structure embedding (Li et al., 2019). Compared with learning-free methods, learnable features have shown more effectiveness. Another advantage of using neural networks is that the graph structure information can be readily embedded into unary node features, as such the classic NP-hard QAP in fact can degenerate into the linear assignment problem, which can be readily solved by existing back-end solvers

---
*Correspondence author is Junchi Yan. The work was in part supported by National Key Research and Development Program of China (2020AAA0107600), National Natural Science Foundation of China (62222607), Shanghai Municipal Science and Technology Major Project (2021SHZDZX0102), and Huawei Technologies.

in polynomial time. Perhaps for this reason, existing deep GM methods (Wang et al., 2019b; Fey et al., 2020a; Zhao et al., 2021; Gao et al., 2021) mostly focus on the front-end learning, basically by supervised learning using manually labeled node correspondence as ground truth. While the back-end solver is relatively little considered for learning in literature: the authors simply combine their front-end feature extractors with some traditional combinatorial solvers e.g. Sinkhorn (Cuturi, 2013), which means they hardly utilize deep learning to improve the back-end solvers.

Note that *outliers* in this paper refer to the common setting in literature (Yang et al., 2015): namely the spurious nodes that cannot find their correspondence in the opposite graph for matching, which are also ubiquitous in real-world matching scenarios. While *inliers* are those having correspondences. More specifically, we assume the most general and challenging case that outliers could exist in both two input graphs, which is in contrast to the majority of works (Zanfir et al., 2018; Wang et al., 2021a) that assume there is at most one graph containing outliers. Though there is a line of works for image matching effectively dismissing the outliers (Fischler & Bolles, 1981; Ma et al., 2020), these works are basically based on specific pose and motion models in vision, which may not always be available for general GM tasks. Moreover, as mentioned above, existing deep GM works are all supervised (or based on supervised learning modules), while in the real world, the labeling is costly and even almost impossible to obtain for large-scale QAP instances in practice.

Towards practical and robust graph matching learning, in the absence of labels and in the presence of outliers (in both input graphs), we propose a reinforcement learning (RL) method for graph matching namely **RGM**, especially for its most general QAP formulation. In particular, RL is conceptually well suited for its label-free nature and flexibility in finding the node correspondence by sequential decision making, which provides a direct way of avoiding outlier over-matching by an early stopping. In contrast, in existing deep GM works, matching is performed in one shot which incurs coupling of the inliers and outliers, and it lacks an explicit way to distinguish outliers. Therefore, we specifically devise a so-called revocable deep reinforcement learning framework to allow small mistakes over the matching procedure, and the current action is revocable to research a better node correspondence based on up-to-date environment information. Our revocable framework is shown cost-effective and empirically outperforms the existing popular techniques for refining the local decision making e.g. Local Rewrite (Chen & Tian, 2019). Moreover, since the standard GM objective refers to maximizing the affinity score between matched nodes, it causes the over-matching issue i.e. the outliers are also incorporated for matching to increase the overall score. To address this issue, we propose to regularize the affinity score such that it discourages unwanted matchings by assigning a negative score to those pairs. Intuitively, the RL agent will naturally stop matching spurious outliers as the objective score will otherwise decrease. With the help of the revocable framework and affinity regularization, our RGM shows promising performances in various experiments. For more clearance, we compare our RGM with most of the existing GM methods in Table 5. Due to the space limit, we place it in the appendix (A.1), where we add a more detailed discussion of the comparison of RGM and existing works. To sum up, the highlights and contributions of our work are:

1) We propose RGM that sequentially selects the node correspondences from two graphs, in contrast to the majority of existing works that obtain the whole matching in one shot. Accordingly, our approach can naturally handle the case for partial matching (due to outliers) by early stopping.

2) Specifically, we first devise a revocable approach to select the possible node correspondence, whose mechanism is adapted to the unlabeled graph data with the affinity score as the reward. To the best of our knowledge, this is the first attempt to successfully adapt RL to graph matching.

3) For avoiding matching the outliers, we develop a regularization to the affinity score, making the solver no longer pursue to match as many nodes as possible. To our best knowledge, this is also the first work for regularizing the affinity score to avoid over-matching among outliers.

4) On synthetic datasets, Willow Object dataset, Pascal VOC dataset, and QAPLIB datasets, RGM shows competitive performance compared with both learning-free and learning-based baselines. Note that RGM focuses on learning the back-end solver and hence it is orthogonal to many existing front-end feature learning based GM methods, which can further boost the front-end learning solvers' performance as shown in our experiments.

## 2 RELATED WORKS

**Graph matching.** It aims to find the node correspondence by considering both the node features as well as the edge attributes, which is known as NP-hard in its general form (Loiola et al., 2007). Classic methods mainly resort to different optimization heuristics e.g. random walk (Cho et al., 2010), spectral matching (Leordeanu & Hebert, 2005), path-following (Zhou & Torre, 2016), graduated assignment (Gold et al., 1996), to SDP relaxation (Schellewald & Schnörr, 2005).

**Deep learning of GM.** Since the seminal work (Zanfir et al., 2018), deep neural networks have been devised for graph matching (Yan et al., 2020). Among the deep GM methods, we generally discuss two representative lines of research as follow. The first line of works (Wang et al., 2019b; Yu et al., 2020) apply CNNs or/and GNNs for learning the node and structure features. By using node embedding, the problem degrades into linear assignment that can be optimally solved by the Sinkhorn network (Cuturi, 2013) to fulfill double-stochasticity which is non-learnable. Instead, another line of studies (Wang et al., 2020c) follow the general Lawler's QAP form exactly, and the problem becomes combinatorial selecting of nodes on the association graphs. Yet all the above models adopt supervised learning and there is little reported success using RL to solve GM despite their label-free advantage and popularity in solving other combinatorial problems.

**Matching against outliers.** Matching against outliers has been a long standing task especially for vision, and seminal works date back to RANSAC (Fischler & Bolles, 1981). There are efforts (Torresani et al., 2012; Yang et al., 2015; Yi et al., 2018; Zhang et al., 2019; Liu et al., 2020) in exploring the specific problem structure and clues in terms of spatial and geometry coherence to achieve robust matching. While we focus on the general setting of GM without using additional assumptions or parametric transform models, the relevant pair-wise graph matching works are relatively less crowded. Wang et al. (2019a) utilizes the strategy based on domain adaptation, which removes the outliers by data normalization module. A heuristic max-pooling strategy is developed in Cho et al. (2014) for dismissing excessive outliers. Wang et al. (2020a) proposes to suppress the matching of outliers by assigning zero-valued vectors to the potential outliers. However, all the above methods are learning-free and the trending learning-based solvers to our best knowledge, have not addressed the outlier problems explicitly except for adding dummy nodes (Wang et al., 2021a).

**Reinforcement learning for combinatorial optimization.** There is growing interest in using RL in solving combinatorial optimization problems (Bengio et al., 2021), such as traveling salesman problem (Khalil et al., 2017), vehicle routing problem (Nazari et al., 2018), job scheduling problem (Chen & Tian, 2019), maximal common subgraph (Bai et al., 2021), The main challenges of these approaches are designing suitable problem representation and tuning reinforcement learning algorithms. For combinatorial optimization problems on single graph, pointer networks (Vinyals et al., 2015) and GNN (Kipf & Welling, 2017) are the most widely used representations. However, for graph matching, there are two graphs for input and the agent needs to pick a node from each of the two graphs every step. To our best knowledge, GM has not been (successfully) addressed by RL.

## 3 PRELIMINARIES

In this paper, we mainly focus on two graph matching i.e. pairwise graph matching, which is also known as pairwise graph matching. Specifically, we consider a more difficult situation, where there are some outliers in both two graphs, and we want to match the similar inliers while ignoring outliers.

Given two weighted graphs $G^1$ and $G^2$, we aim to find the matching between their nodes such that the affinity score is maximized. We use $V^1$ and $V^2$ to represent the nodes of graph $G^1$ and $G^2$. We suppose that $|V^1| = n_1, |V^2| = n_2$, and **there can be outliers in $G^1$, $G^2$, or both**. $E^1$ and $E^2$ denote the edge attributes of graph $G^1$ and $G^2$. The affinities of pairwise graph matching include the first order (node) affinities and the second order (edge) affinities. Generally, the graph matching problem can be regarded as Lawler's Quadratic Assignment Problem (Lawler, 1963):

$$J(\mathbf{X}) = \text{vec}(\mathbf{X})^\top \mathbf{K} \, \text{vec}(\mathbf{X}),$$
$$s.t. \quad \mathbf{X} \in \{0,1\}^{n_1 \times n_2}, \ \mathbf{X}\mathbf{1}_{n_2} \le \mathbf{1}_{n_1}, \ \mathbf{X}^\top \mathbf{1}_{n_1} \le \mathbf{1}_{n_2} \tag{1}$$

where $\mathbf{X}$ is the permutation matrix of which the element is 0 or 1, $\mathbf{X}(i,a) = 1$ denotes node $i$ in graph $G^1$ is matched with node $a$ in graph $G^2$. The operator $\text{vec}(\cdot)$ means column-vectorization. $\mathbf{K} \in \mathbb{R}^{n_1 n_2 \times n_1 n_2}$ is the affinity matrix. For node $i$ in $G^1$ and node $a$ in $G^2$ the node-to-node affinity is encoded by the diagonal element $\mathbf{K}(ia, ia)$, while for edge $ij$ in $G^1$ and edge $ab$ in $G^2$ the edge-to-edge affinity is encoded by the off-diagonal element $\mathbf{K}(ia, jb)$. Assuming $i, a$ both start from 0,

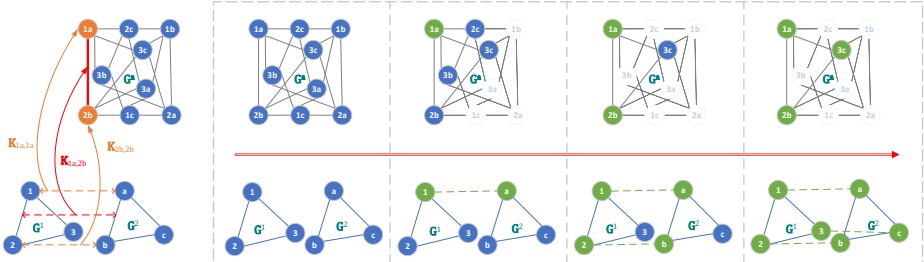

Figure 1: Left: the association graph ($G^a$ on the top) can be derived from the raw input graphs ($G^1, G^2$ at the bottom). Right: the matching process of our RL procedure: the blue vertices denote available vertices, the green vertices denote selected (matched) vertices, and the blurred vertices denote the unavailable vertices. The agent selects "1a", "2b", and "3c" progressively on $G^a$.

the index $ia$ means $i \times n_2 + a$. The objective is to maximize the sum of both first order and second order affinity score $J(\mathbf{X})$ given the affinity matrix $\mathbf{K}$ by finding an optimal permutation $\mathbf{X}$.

Graph matching involves (at least) two input graphs. Instead of directly working on two individual graphs which can disclose raw data information and might be sensitive, in this paper, we first construct the association graph of the pairwise graphs as the input representation (Leordeanu & Hebert, 2005; Wang et al., 2021a). Specifically, we construct an association graph $G^a = (V^a, E^a)$ from the original pairwise graph $G^1$ and $G^2$, with the help of the affinity matrix $\mathbf{K}$. We merge each two nodes $(v_i, v_a) \in V^1 \times V^2$ as a vertex $v_p \in V^a$. We can see that the association graph contains $|V^a| = n_1 \times n_2$ vertices[1].There exists an edge for every two vertices as long as they do not contain the same node from the original pairwise graphs, so every vertex is connected to $(n_1 - 1) \times (n_2 - 1)$ edges. There exist both vertex weights $w(v_p)$ and edge weights $w(v_p, v_q)$ in the association graph, where the vertex and edge weights denote the first and second order affinities:

$$\begin{aligned} \mathbf{F}(p, p) = w(v_p) = \mathbf{K}(ia, ia), \text{ where } p = ia \\ \mathbf{W}(p, q) = w(v_p, v_q) = \mathbf{K}(ia, jb), \text{ where } p = ia, q = jb \end{aligned} \quad (2)$$

where the vertex index $p$ in the association graph $G^a$ means a combination of the indices $i$ and $a$ in the original pairwise graphs $G^1$ and $G^2$. $\mathbf{F}, \mathbf{W} \in \mathbb{R}^{n_1 n_2 \times n_1 n_2}$ are the weight matrices that contain the vertex weights and edge weights in the association graph, respectively. Fig. 1 shows an example to construct the association graph from the input graphs. In the association graph $G^a$, selecting a vertex $p$ in the association graph equals to matching nodes $i$ and $a$ in the input graphs. Therefore, we can select a set of vertices $\mathbb{U}$ as we choose to match these nodes as the solution. Note that the set of vertices $\mathbb{U}$ in the association graph is equivalent to the permutation matrix $\mathbf{X}$ and can be easily converted, as long as the set $\mathbb{U}$ does not violate the constraint in Eq. 1.

## 4 APPROACH

### 4.1 REINFORCEMENT LEARNING FOR GRAPH MATCHING

We design our RGM based on Double Dueling DQN (D3QN) (Hasselt et al., 2016), as it is widely accepted (Sutton & Barto, 2018) that value based RL algorithms like D3QN can better handle the discrete case, which will be also verified in our ablation study. The pipeline of the training process of RGM is described in Alg. 1. Details of the proposed method are given in Appendix (A.2), including network structure (A.2.1), prioritized experience replay memory (A.2.2) and model updating algorithm (A.2.3). We show the design of state, action, and reward in our RGM:

**1) State.** The state $s$ is the current (partial) solution $\mathbb{U}'$, where $\mathbb{U}'$ is also a set of vertices in the association graph, with $|\mathbb{U}'| \leq min(n_1, n_2)$. The size of $\mathbb{U}'$ increases from 0 at the beginning, and finally partial solution $\mathbb{U}'$ becomes a complete solution $\mathbb{U}$ when the agent decides to stop the episode.

**2) Action.** The action $a$ of our reinforcement learning agent is to select a vertex in the association graph and add it to the current solution $\mathbb{U}'$. By the definition of graph matching, we can not match two nodes in $G^1$ to one same node in $G^2$ and vice versa. Therefore, in our basic RL framework, we can only select the vertices in the available vertex set. Take Fig. 1 for an example, once we select the vertex "1a", it means we have matched node "1" in $G^1$ and node "a" in $G^2$. Then, we can not

---

[1]For clarification, this paper uses "node" for the raw graphs and "vertex" for the association graph.

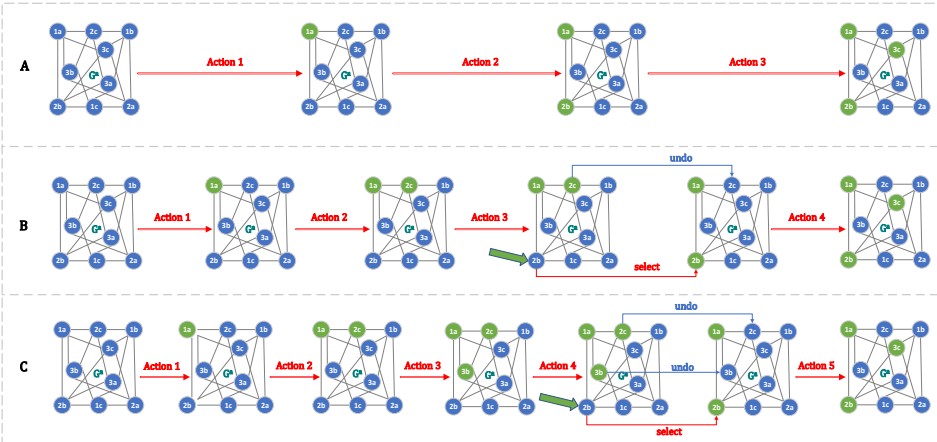

Figure 2: The different cases of our proposed revocable RGM in several situations.

match node "1" to node "b" or "c" later, which means we can not select vertices "1b" or "1c". Given partial solution $\mathbb{U}'$, the available vertices set $\mathbb{V}$ is written as:

$$\mathbb{V} = \{v \mid \mathbf{A}(v, v') = 1, \ \forall \ v' \in \mathbb{U}', \ v \in V^a\} \quad (3)$$

where $V^a$ is the vertices in the association graph, whose adjacency matrix (0 for no edge connected and 1 for existing edge connected) is $\mathbf{A}$. Eq. 3 holds since two vertices are connected if they do not contain the same node from the input graphs. If a vertex is connected to all vertices in $\mathbb{U}'$, then it has no conflict. Given the available set, we mask all unavailable vertices in the association graph to make sure that the agent can not select them, as illustrated by the blurred vertices in Fig. 1 .

Then, the action is to pick a node $v$ from the available vertices set $\mathbb{V}$: $\mathbb{U}_{old} \xrightarrow{v \in \mathbb{V}} \mathbb{U}_{new}$, where $\mathbb{U}_{old}$ is the old partial solution, and $\mathbb{U}_{new}$ is the new partial solution after an action.

It is worth noting that, the requirement that the agent needs to select the vertex from the available set only exists in our basic RL framework. **In our later proposed revocable RL framework, the agent can select any vertex without constraint.**

**3) Reward.** We define the reward $r$ as the improvement of the objective score between the old partial solution and the new partial solution after executing an action.

### 4.2 THE REVOCABLE ACTION MECHANISM

So far we have presented a basic RL framework. In such a vanilla form, it can not undo the actions that have been executed. In other words, the agent can not "regret", which means the agent has no chance to adjust if the agent makes a wrong decision and the error may accumulate until obtaining a disastrous solution. To strike a reasonable trade-off between efficiency and efficacy, we develop a mechanism to allow the agent to re-select the vertex on the association in one revocable step.

To allow the agent to modify the decisions made before, we design a new revocable RL framework. We remove the available set used before, and **the agent is free to choose any vertex, even if it is in conflict.** Then, we modify the strategy in our RL environment. If the environment receives a new vertex from the agent's action that is in conflict with currently selected vertices, the environment will remove one or two vertices that are in conflict with the new coming vertex, and then add the new vertex to the current solution. Our proposed revocable RL framework is illustrated in Fig. 2.

As Fig. 2 shows, the pipelines of our revocable RGM are a little different from our basic framework. The available set in the basic RGM does not exist if the revocable mode is on. Pipeline (A) shows a simple situation that the agent matched the vertices directly without any reverse operation. In pipeline (B), we suppose the agent chooses the vertex "2c" for the second action, which is not a good choice. When choosing the third action, the agent realizes that it made a mistake by selecting "2c", therefore, the agent chooses "2b" to fix this mistake. Then, this action is passed to the environment, and the environment reverts "2c" and selects "2b" instead. In other words, the agent can reverse "2c" to "2b". In pipeline (C), we show another revocable situation. Suppose that the agent selects "2c" and "3b" as its second and third actions, and acquires a complete matching solution. However, it turns out the matching solution ("1a", "2c", and "3b") is not as good as expected. To roll this

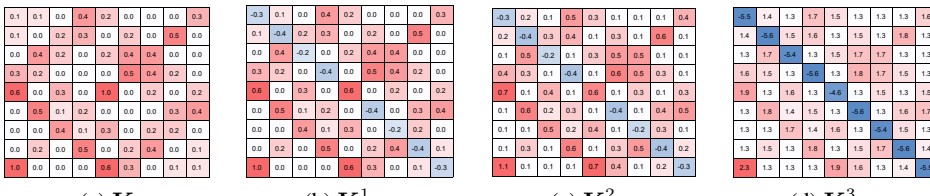

|   (a) $\mathbf{K}$   |   (b) $\mathbf{K}_{reg}^1$   |   (c) $\mathbf{K}_{reg}^2$   |   (d) $\mathbf{K}_{reg}^3$   |

Figure 3: Examples of the original affinity matrix and the regularized affinity matrix: (a) original affinity matrix $\mathbf{K}$; (b)(c)(d) regularized affinity matrix $\mathbf{K}_{reg}$ by quadratic approximation, with three different regularization functions, of which some values become negative.

situation back, the agent can select "2b" as the next action. By selecting "2b", the environment will release the vertices "2c" and "3b", and then select "2b". Finally, the agent chooses "3c" as the last action, and decides to end the episode with the matching solution ("1a", "2b", and "3c").

Above all, our proposed revocable reinforcement learning framework allows the agent to make better decisions by giving the opportunity to turn the way back. From now on in the paper, the default setting of RGM contains the revocable action framework. For further discussion, especially the difference from other similar frameworks (especially the popular Local Rewrite (Chen & Tian, 2019) and ECO-DQN (Barrett et al., 2020)), please refer to the appendix(A.2.4).

### 4.3 OUTLIER-ROBUST GRAPH MATCHING

For practical GM, outliers are common in both the input pairwise graphs. In general, the solution is supposed to contain only the inlier correspondences without outliers. However, most existing GM methods are designed to match all keypoints whether they are inliers or outliers. This design is based on pursuing the highest objective score, and matching outliers also can increase the objective score to some extent. We believe that the outliers should not be matched in any sense. Therefore, we propose two strategies to guide the agent to match the inliers only and ignore the outliers.

#### 4.3.1 INLIER COUNT INFORMATION EXPLORATION

Our first strategy is to inform the agent of the amount of common inliers. Similar input settings can also be found in learning-free outlier-robust GM methods (Yang et al., 2017; Wang et al., 2020a). Given the exact number of inliers $n_i$, the sequential matching of RGM can be readily stopped when the size of the current solution $|\mathbb{U}'| = n_i$, leaving the remaining $n_1 - n_i$ and $n_2 - n_i$ nodes as outliers.

#### 4.3.2 AFFINITY REGULARIZATION VIA QUADRATIC APPROXIMATION

Our second strategy is to regularize the affinity score (i.e. objective score). As mentioned before, existing GM methods tend to match all keypoints including outliers to make the affinity score as high as possible. Therefore, one straightforward idea is to regularize the affinity score which exerts penalization on over-matching terms to balance the effect of the outliers.

In this paper, we propose to regularize the original affinity score that blindly adds up all the node/edge correspondence affinity values including the outliers. In other words, we aim to design an inlier-aware *regularized affinity score* denoted by $J_{reg}(\mathbf{X})$ to dismiss the effect of outliers in affinity score computing, as the affinity score from the inlier matching is more meaningful. Specifically, the regularization is devised as a function w.r.t. the number of matched keypoints in $\mathbf{X}$, as denoted by $f(\|\mathrm{vec}(\mathbf{X})\|_1)$, which is multiplied on the original affinity score as follows:

$$J_{reg}(\mathbf{X}) = \mathrm{vec}(\mathbf{X})^\top \mathbf{K} \mathrm{vec}(\mathbf{X}) \cdot f(\|\mathrm{vec}(\mathbf{X})\|_1) \tag{4}$$

In general, $f(\|\mathrm{vec}(\mathbf{X})\|_1)$ shall become smaller when there are more matched keypoints to suppress spurious outliers. For the effectiveness, in this paper we choose the following three functions without loss of generality: $f_1(n) = \frac{3\max(n_1,n_2)-n}{3\max(n_1,n_2)}$, $f_2(n) = \frac{1+n}{1+3n}$, $f_3(n) = \frac{1}{n^2}$.

However, the above formula is unfriendly for learning in RGM as our core network GNN can only accept the form of the affinity matrix $\mathbf{K}$ namely the association graph as input, while the impact of $f(\|\mathrm{vec}(\mathbf{X})\|_1)$ can not be delivered to the GNN. To fill this gap, in the following, we further propose a technique to transform Eq. 4 into a standard QAP formulation, and construct the regularized affinity matrix $\mathbf{K}_{reg}$ as the input for the GNN, instead of the original affinity matrix $\mathbf{K}$.

Recall the original score in Eq. 1 and we further denote $n_x = \|\mathrm{vec}(\mathbf{X})\|_1$. Eq. 4 can be rewritten as:

$$J_{reg}(\mathbf{X}) = \mathrm{vec}(\mathbf{X})^\top \mathbf{K} \mathrm{vec}(\mathbf{X}) - J \cdot (1 - f(n_x)) \approx \mathrm{vec}(\mathbf{X})^\top \mathbf{K} \mathrm{vec}(\mathbf{X}) - J \cdot g(n_x) \tag{5}$$

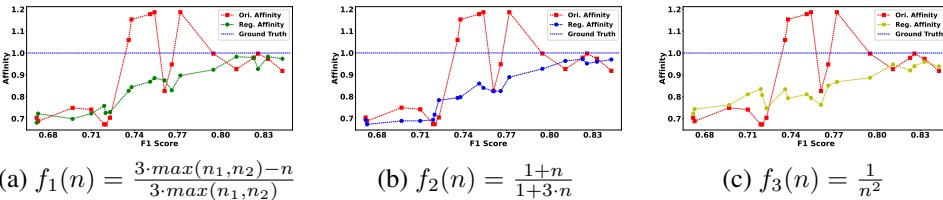

$$\text{(a) } f_1(n) = \frac{3 \cdot max(n_1, n_2) - n}{3 \cdot max(n_1, n_2)} \qquad \text{(b) } f_2(n) = \frac{1+n}{1+3 \cdot n} \qquad \text{(c) } f_3(n) = \frac{1}{n^2}$$

Figure 4: The empirical relation between F1 and original/regularized affinity scores. The affinity score of the original affinity matrix is calculated by $\frac{\text{vec}(\mathbf{X}^{pred})^\top \mathbf{K} \text{vec}(\mathbf{X}^{pred})}{\text{vec}(\mathbf{X}^{gt})^\top \mathbf{K} \text{vec}(\mathbf{X}^{gt})}$, and affinity score of the regularized affinity matrix is $\frac{\text{vec}(\mathbf{X}^{pred})^\top \mathbf{K}_{reg} \text{vec}(\mathbf{X}^{pred})}{\text{vec}(\mathbf{X}^{gt})^\top \mathbf{K}_{reg} \text{vec}(\mathbf{X}^{gt})}$. The score of ground truth is constantly 1.

In the above formula, it is worth noting that to make the above formula a quadratic one, we introduce a quadratic function: $g(n) = an^2 + bn + c$, which is used to approximate the term $g(n) \approx 1 - f(n)$, in a way of least square fitting to determine the unknown coefficients $a, b, c$. Technically the fitting involves sampling $n$ in a certain range according to the prior of matching problem at hand, e.g. $n \in \{7, 8, 9, 10, 11, 12\}$. Then, $g(n)$ can be expanded by: ($\mathbf{K}_A = \mathbf{1}$ (all-one matrix) and $\mathbf{K}_B = \mathbf{I}$)

$$
\begin{aligned}
g(n_x) &= a \sum_{ijkl} \mathbf{X}(i,j) \cdot \mathbf{X}(k,l) + b \sum_{ij} \mathbf{X}(i,j) \cdot \mathbf{X}(i,j) + c \\
&= a \cdot \text{vec}(\mathbf{X})^\top \mathbf{K}_A \text{vec}(\mathbf{X}) + b \cdot \text{vec}(\mathbf{X})^\top \mathbf{K}_B \text{vec}(\mathbf{X}) + c \\
&= \text{vec}(\mathbf{X})^\top (a \mathbf{K}_A + b \mathbf{K}_B) \text{vec}(\mathbf{X}) + c
\end{aligned}
\tag{6}
$$

During the iteratively problem solving in RGM, we note that $J$ changes relatively slowly between two consecutive iterations, and thus we treat it as a constant during iteration. Based on this observation, we then try to approximately convert $J_{reg}$ into a QAP formulation as follows:

$$J_{reg}(\mathbf{X}) \approx \text{vec}(\mathbf{X})^\top (\mathbf{K} - aJ \cdot \mathbf{K}_A + bJ \cdot \mathbf{K}_B) \text{vec}(\mathbf{X}) \tag{7}$$

which is friendly to GNN learning. Let $\mathbf{K}_{reg} = \mathbf{K} - aJ \cdot \mathbf{K}_A - bJ \cdot \mathbf{K}_B$ and Eq. 5 becomes:

$$\mathbf{X} = \arg\max_{\mathbf{X}} \ \text{vec}(\mathbf{X})^\top \mathbf{K} \text{vec}(\mathbf{X}) \cdot f(\|\text{vec}(\mathbf{X})\|_1) \approx \arg\max_{\mathbf{X}} \text{vec}(\mathbf{X})^\top \mathbf{K}_{reg} \text{vec}(\mathbf{X}) \tag{8}$$

In this way, we now can input the regularized affinity matrix $\mathbf{K}_{reg}$ to the GNN, to better learn the impact of the regularization term $f(\|\text{vec}(\mathbf{X})\|_1)$. Fig. 3 shows the results of the regularized affinity matrix. Notably, some values in the regularized affinity $\mathbf{K}_{reg}$ become negative. Intuitively, the negative elements in the affinity matrix denote that the affinity score maximization reward may no longer pursue to match as many keypoints pairs as possible, which prevents the agent from picking up outliers to some extent. Besides, most traditional graph matching solvers (Cho et al., 2010; Egozi et al., 2013) assume the affinity matrix is non-negative, while our RGM has no such restriction.

Fig. 4 illustrates the effectiveness of *regularized affinity score*, with three different regularization functions $\{f_i(n)\}_{i=1,2,3}$. We construct a set of permutation solutions by GM solver, and calculate the F1 score and affinity score of the original affinity matrix and regularized affinity matrix. As one can see, the affinity score of the original affinity matrix fluctuates a lot with the increase of the F1 score. Besides, there are some cases where original affinity is larger than ground truth matching. In contrast, the *regularized affinity score* of the regularized affinity matrix is more stable and overall nearly positively correlates to the F1 score. It proves that the *regularized affinity score* is a better optimization objective that is consistent with matching accuracy (F1 score).

This approximate optimization process can be merged into our revocable framework as shown in Alg. 1. In the sequential decision making process, at every step when the agent needs to select the next vertex, we make an approximate optimization based on the current solution, and get the regularized affinity matrix $\mathbf{K}_{reg}$. Then, we feed the regularized affinity matrix $\mathbf{K}_{reg}$ instead of the original affinity matrix $\mathbf{K}$ to the GNN. The above introduced regularization technique can be readily adopted in our sequential node matching scheme. In contrast, it is nontrivial for them to be integrated with existing deep GM methods which are mainly performed in one shot for the whole matching.

## 5 EXPERIMENTS

We perform experiments on the standard benchmarks including image datasets (synthetic image, Willow Object, Pascal VOC) as well as pure combinatorial optimization instances (QAPLIB), as

Table 1: Average performance w.r.t F1 score and objective score (the higher the better) in the Willow Object dataset with respect to different numbers of randomly added outliers given the 10 inliers. "AR": affinity regularization, "IC": inlier count information, "w/o rev.": without the revocable.

| | Outlier # | 1 | | 2 | | 3 | | 4 | | 5 | | 6 | |
|---|---|---|---|---|---|---|---|---|---|---|---|---|---|
| | Method | F1 | Obj | F1 | Obj | F1 | Obj | F1 | Obj | F1 | Obj | F1 | Obj |
| Learning-free | RRWM (Cho et al., 2010) | 73.90% | 0.9062 | 71.57% | 0.8401 | 66.03% | 0.7989 | 61.27% | 0.7595 | 56.08% | 0.7077 | 52.21% | 0.6836 |
| | GAGM (Gold et al., 1996) | 69.17% | 0.8488 | 62.99% | 0.7657 | 60.46% | 0.7365 | 57.00% | 0.7133 | 53.85% | 0.6675 | 52.71% | 0.6658 |
| | IPFP (Leordeanu et al., 2009) | 80.89% | 0.9189 | 73.88% | 0.8436 | 69.41% | 0.8063 | 64.07% | 0.7896 | 58.33% | 0.7113 | 55.36% | 0.6893 |
| | PSM (Egozi et al., 2013) | 84.33% | 0.9047 | 71.35% | 0.8302 | 66.46% | 0.7914 | 60.17% | 0.7453 | 54.56% | 0.6862 | 51.06% | 0.6534 |
| | GNCCP (Liu et al., 2012) | 85.81% | 0.9237 | 77.82% | 0.8564 | 71.07% | 0.8164 | 65.47% | 0.7737 | 59.30% | 0.7125 | 56.17% | 0.6891 |
| | BPF (Wang et al., 2018) | 85.89% | 0.9239 | 78.18% | 0.8566 | 71.26% | 0.8131 | 65.74% | 0.7736 | 59.47% | 0.7119 | 56.31% | 0.6901 |
| | ZACR (Wang et al., 2020a) | 81.71% | 0.9121 | 74.68% | 0.8452 | 67.98% | 0.7824 | 63.77% | 0.7514 | 57.73% | 0.6997 | 54.75% | 0.6821 |
| Supervised | GMN (Zanfir et al., 2018) | 75.52% | - | 72.35% | - | 65.64% | - | 56.64% | - | 55.41% | - | 51.95% | - |
| | PCA (Wang et al., 2019b) | 79.78% | - | 74.01% | - | 67.71% | - | 59.41% | - | 57.41% | - | 52.31% | - |
| | LCS (Wang et al., 2020c) | 86.27% | - | 77.68% | - | 70.49% | - | 66.79% | - | 59.32% | - | 55.61% | - |
| | NGM (Wang et al., 2021a) | 81.24% | - | 74.37% | - | 68.31% | - | 61.93% | - | 56.87% | - | 53.32% | - |
| | BBGM (Rolínek et al., 2020) | 87.21% | - | 79.84% | - | 73.41% | - | 69.09% | - | 61.79% | - | 58.07% | - |
| | NGM-v2 (Wang et al., 2021a) | 86.84% | - | 78.90% | - | 73.23% | - | 68.96% | - | 60.52% | - | 56.73% | - |
| RL | RGM + AR | 85.63% | **0.9664** | 80.09% | **0.9543** | 75.65% | **0.9288** | 71.20% | **0.9111** | 64.11% | **0.8845** | 62.10% | **0.8682** |
| | RGM + IC | 86.18% | 0.9426 | 80.11% | 0.9235 | 76.72% | 0.8860 | 71.78% | 0.8673 | 65.26% | 0.8437 | 64.27% | 0.8131 |
| | RGM + AR + IC | **87.68%** | 0.9605 | **80.37%** | 0.9443 | **77.57%** | 0.9122 | **73.77%** | 0.8919 | **67.49%** | 0.8603 | **65.10%** | 0.8397 |
| | RGM + AR w/o rev. | 84.97% | 0.9239 | 78.95% | 0.8994 | 72.19% | 0.8702 | 68.32% | 0.8562 | 61.50% | 0.8052 | 59.00% | 0.7900 |
| | RGM + IC w/o rev. | 85.91% | 0.9111 | 79.10% | 0.8735 | 72.42% | 0.8360 | 69.07% | 0.8173 | 62.76% | 0.7737 | 61.07% | 0.7431 |
| | RGM + AR + IC w/o rev. | 86.41% | 0.9274 | 79.76% | 0.8936 | 73.92% | 0.8618 | 70.12% | 0.8411 | 63.92% | 0.7988 | 61.26% | 0.7574 |

Table 2: Sensitivity test by using inexact inlier count $n_i$ ranging from 8 to 13, instead of the ground truth 10 on Willow Object. Experiments are conducted with 10 inliers and **3** outliers.

| $n_i$ | 8 | | 9 | | 10 (ground truth) | | 11 | | 12 | | 13 | |
|---|---|---|---|---|---|---|---|---|---|---|---|---|
| Method | F1 | Obj | F1 | Obj | F1 | Obj | F1 | Obj | F1 | Obj | F1 | Obj |
| BPF (Wang et al., 2018) | 71.26% | 0.8131 | 71.26% | 0.8131 | 71.26% | 0.8131 | 71.26% | 0.8131 | 71.26% | 0.8131 | 71.26% | 0.8131 |
| BBGM (Rolínek et al., 2020) | 73.41% | - | 73.41% | - | 73.41% | - | 73.41% | - | 73.41% | - | 73.41% | - |
| RGM + AR | **75.65%** | **0.9288** | **75.65%** | **0.9288** | 75.65% | **0.9288** | 75.65% | **0.9288** | 75.65% | 0.9288 | **75.65%** | 0.9288 |
| RGM + IC | 70.78% | 0.8075 | 73.78% | 0.8698 | 76.72% | 0.8860 | 73.61% | 0.9163 | 71.40% | **0.9433** | 66.96% | **0.9553** |
| RGM + AR +IC | 71.94% | 0.8639 | 74.87% | 0.8967 | **77.57%** | 0.9122 | **75.73%** | 0.9184 | 74.78% | 0.9244 | 73.89% | 0.9281 |

the latter is suited for back-end solver. Due to the page limit, the detailed experiment protocols are introduced in the appendix (A.3), including hyperparameter settings (A.3.1), evaluation metrics (A.3.2), compared methods (A.3.3), dataset settings (A.3.4), and the train/test protocols (A.3.5).

## 5.1 EXPERIMENTS ON WILLOW OBJECT DATASET WITH OUTLIERS

**1) Performance over Different Amounts of Outliers.** We conduct more experiments with respect to different amounts of outliers. In these experiments, the number of inliers is fixed at 10 while the number of outliers varies from 1 to 6. The experiment setup follows Jiang et al. (2021), and the evaluation is performed on all five categories. For the training and testing process, we follow the data split rate in BBGM (Rolínek et al., 2020) in all experiments. Table 1 shows the performance of our method and baselines in the average of all five classes on the Willow Object dataset with respect to different amounts of outliers (from 1 to 6). For our methods at the bottom, "RGM + AR" denotes RGM with affinity regularization, "RGM + IC" denotes RGM with inlier count information, and "RGM + AR + IC" denotes RGM with both. For ablation, RGM without the revocable mechanism is shown in the last three columns, denoted by "w/o rev.". Regularized affinity and inlier count information are two ways to handle the outliers introduced in Sec. 4.3. RGM reaches the best results over all five classes in this dataset, and gets a 4% improvement in average compared to the best baseline BBGM. For the variants of our methods, "RGM + AR + IC" shows the best performance, while "RGM + AR" requires no extra information and still outperforms all the baselines.

**2) Sensitivity to the Input of Inlier Count.** Since in reality, the inlier count information might not be always accurate, we test our methods' robustness against the inexact number of inliers. We re-conduct experiments, of which each image contains 10 inliers and 3 outliers. We modify the inlier count info $n_i$ to (8 - 13) instead of ground truth 10, as the input to "RGM + IC" and "RGM + AR + IC". The results in Table 2 show that the performances of these two methods tend to downgrade given incorrect information, as expected, but they can still outperform BBGM in cases when the information is only a little biased. Meanwhile, the regularized affinity matrix can also enhance the robustness. As a result, our RGM can still work well even the inlier count information is inaccurate.

## 5.2 EXPERIMENTS ON PASCAL VOC

Table 3 reports the results on the Pascal VOC. We also apply the train/test split rate as mentioned in Sec. A.3.5. We show the average performance of every 20 classes in this dataset, and we can see that RGM outperforms all baselines on Pascal VOC. Specifically, the baselines on the left side are

Table 3: Average performance over all classes on Pascal VOC. "w/ label" denotes requiring a label.

| | RRWM | GAGM | IPFP | PSM | GNCCP | BPF | RGM | GMN | PCA | LCS | BBGM | NGM | NGM-v2 | NGM + RGM | NGM-v2 + RGM |
|---|---|---|---|---|---|---|---|---|---|---|---|---|---|---|---|
| F1 | 48.58% | 53.52% | 41.94% | 53.72% | 49.98% | 54.70% | 60.19% | 55.30% | 64.80% | 68.50% | 79.00% | 64.10% | 80.10% | 68.04% | **81.88%** |
| Obj | 1.017 | 1.019 | 1.017 | 1.019 | 1.018 | 1.020 | **1.040** | - | - | - | - | - | - | 1.015 | 1.011 |
| w/ label | × | × | × | × | × | × | × | √ | √ | √ | √ | √ | √ | √ | √ |

Table 4: Performance gap with the optimal (%) (the lower the better) on QAPLIB, the mean/max/min gaps are reported for each class. The mean performance over all classes and the inference time (s) per instance are reported. The number in the bracket is the size of instances in each category.

| | bur (26) | | | chr (12 - 25) | | | esc (16 - 64) | | | had (12 - 20) | | | kra (30 - 32) | | | lipa (20 - 60) | | | nug (12 - 30) | | | rou (12 - 30) | | |
|---|---|---|---|---|---|---|---|---|---|---|---|---|---|---|---|---|---|---|---|---|---|---|---|---|
| | mean | min | max | mean | min | max | mean | min | max | mean | min | max | mean | min | max | mean | min | max | mean | min | max | mean | min | max |
| SM | 22.3 | 20.3 | 24.9 | 460.1 | 144.6 | 869.1 | 301.6 | 0.0 | 3300.0 | 17.4 | 14.7 | 21.5 | 65.3 | 63.8 | 67.3 | 19.0 | 3.8 | 34.8 | 45.5 | 34.2 | 64.0 | 35.8 | 30.9 | 38.2 |
| RRWM | 23.1 | 19.3 | 27.3 | 616.0 | 120.5 | 1346.3 | 63.9 | 0.0 | 200.0 | 25.1 | 22.1 | 28.3 | 58.8 | 53.9 | 67.7 | 20.9 | 3.6 | 41.2 | 67.8 | 52.6 | 79.6 | 51.2 | 39.3 | 60.1 |
| SK-JA | 4.7 | 2.8 | 6.2 | **38.5** | **0.0** | **186.1** | 364.8 | 0.0 | 2200.0 | 25.8 | 6.9 | 100.0 | 41.4 | 38.9 | 44.4 | **0.0** | **0.0** | **0.0** | 25.3 | 10.9 | 100.0 | 13.7 | 10.3 | 17.4 |
| NGM | **3.4** | **2.8** | **4.4** | 121.3 | 45.4 | 251.9 | 126.7 | 0.0 | 200.0 | 8.2 | 6.0 | 11.6 | 31.6 | 28.7 | 36.8 | 16.2 | 3.6 | 29.4 | 21.0 | 14.0 | 28.5 | 30.9 | 23.7 | 36.3 |
| RGM | 7.1 | 4.5 | 9.0 | 112.4 | 23.4 | 361.4 | **32.8** | **0.0** | **141.5** | **6.2** | **1.9** | **9.0** | **15.0** | **10.4** | **20.6** | 13.3 | 3.0 | 23.8 | **9.7** | **6.1** | **12.9** | **13.4** | **7.1** | **16.7** |

| | scr (12 - 20) | | | sko (42 - 64) | | | ste (36) | | | tai (12 - 64) | | | tho (30 - 40) | | | wil (50) | | | Average (12 - 64) | | | Time per instance |
|---|---|---|---|---|---|---|---|---|---|---|---|---|---|---|---|---|---|---|---|---|---|---|
| | mean | min | max | mean | min | max | mean | min | max | mean | min | max | mean | min | max | mean | min | max | mean | min | max | (in seconds) |
| SM | 123.4 | 104.0 | 139.1 | 29.0 | 26.6 | 31.4 | 475.5 | 197.7 | 1013.6 | 180.5 | 21.6 | 1257.9 | 55.0 | 54.0 | 56.0 | 13.8 | 11.7 | 15.9 | 181.2 | 46.9 | 949.9 | **0.01** |
| RRWM | 173.5 | 98.9 | 218.6 | 48.5 | 47.7 | 49.3 | 539.4 | 249.5 | 1117.8 | 197.2 | 26.8 | 1256.7 | 80.6 | 78.2 | 83.0 | 18.2 | 12.5 | 23.8 | 169.5 | 49.5 | 432.9 | 0.15 |
| SK-JA | 48.6 | 44.3 | **55.7** | 18.3 | 16.1 | 20.5 | 120.4 | 72.5 | 200.4 | 25.2 | **1.6** | 107.1 | 32.9 | 30.6 | 35.3 | 8.8 | **6.7** | 10.7 | 93.2 | **9.0** | 497.9 | 563.44 |
| NGM | 55.5 | 41.4 | 66.2 | 25.2 | 22.8 | 27.7 | **101.7** | **57.6** | **172.8** | 61.4 | 18.7 | 352.1 | 27.5 | 24.8 | 30.2 | 10.8 | 8.2 | 11.1 | 62.4 | 17.8 | 129.7 | 15.72 |
| RGM | **45.5** | **30.2** | 56.1 | **10.6** | **9.9** | **11.2** | 134.1 | 69.9 | 237.0 | **17.3** | 11.4 | **28.6** | **20.7** | **12.7** | **28.6** | **8.1** | 7.9 | **8.4** | **35.8** | 10.7 | 101.1 | 75.53 |

label-free methods, which use the same input as RGM thus leading to a fair comparison. In other words, the left side methods are the pure back-end solvers that pursue the highest objective score. We can see that our RGM reaches the highest objective score 1.040, which is clearly greater than 1. However, even with this high objective score, the matching accuracy of RGM is still unsatisfactory.

Then, we conduct experiments with methods on the right side that requires label as supervision, which means their backend-solvers are trained to reach the higher accuracy instead of the objective score. We combine our RGM with SOTA method NGM and NGM-v2, by using their well-trained model as the guidance of RGM for imitation learning. Then, we can find that RGM can boost the performance of original methods with the modified direction, which expands the usage of our RGM.

## 5.3 EXPERIMENTS ON QAPLIB DATASET

For QAPLIB (Burkard et al., 1997), we use the settings exactly the same as NGM. The results are shown in Table 4, where "esc(16-64)" denotes that the size of class "esc" varies from 16 to 64. The train-test split rate for this dataset is (1 : 1). We calculate the gap between the computed solution and the optimal, and report the average optimal gap (the lower the better). Besides, inference time per instance is listed in the last column. We compare with four existing solvers: SM (Leordeanu & Hebert, 2005), Sinkhorn-JA (Kushinsky et al., 2019), RRWM (Cho et al., 2010) and NGM (Wang et al., 2021a). Note that the NGM is the first that utilizes deep learning to solve QAP which is an emerging topic. It shows that RGM outperforms all the baselines, including the latest solver NGM.

## 5.4 ADDITIONAL EXPERIMENTS

Here we list an index of those experiments in the appendix. If you are interested in several experiments listed below, please refer to Appendix. **(A.4.1) Experiments on the synthetic dataset**: evaluating RGM on the synthetic images. **(A.4.2) Experiments on the different categories of the Willow Object**: results over five categories on the Willow Object dataset. **(A.4.3) Generalization test of RGM**: generalization to different amounts of outliers, similar categories, and different classes of QAP instances. **(A.4.4) Hyperparameter sensitivity study**: testing the sensitivity of the hyperparameters in RGM. **(A.4.5) Seeded graph matching**: graph matching with several initial seeds. **(A.4.6) Revocable action v.s. Local Rewrite**: comparison of RGM and popular Local Rewrite framework (Chen & Tian, 2019). **(A.4.7) Alternative RL backbones**: testing alternative RL backbones instead of D3QN in RGM. **(A.4.8) Limitation analysis of the inconsistency**: analysis of the inconsistency between the affinity score and the F1 score in the aforementioned experiments.

## 6 CONCLUSION

We have presented a deep RL based approach for graph matching, especially in the presence of outliers. The sequential decision scheme allows to natural select the inliers for matching and avoids matching outliers. To our best knowledge, it is the first work for RL of graph matching which can be applied to its general QAP form. We further devise two techniques to improve the robustness. The first is the revocable action mechanism which is shown well suited to our complex constrained search procedure. The other is the affinity regularization based on parametric function fitting, which is shown can effectively refrain the agent from matching outliers when the number of inliers is unknown. Experiments on multiple real-world datasets show the cost-effectiveness of RGM.

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

Table 5: Representative deep GM works. KB's means Koopmans-Beckmann's QAP which is a special form of Lawler's QAP. The appearance feature and structure feature are often modeled by CNN and by GNN, respectively. The affinity model is often relatively simple by a Gaussian kernel or MLP.

| | QAP Form | | Learning Components* | | | | Matching Process | | Learning Protocol | | |
|---|---|---|---|---|---|---|---|---|---|---|---|
| | Lawler's | KB's | Appearance | Structure | Affinity | Back-end Solver | One-shot | Sequential | Supervised | Self-supervised | Reinforcement |
| GMN (Zanfir et al., 2018) | √ | | √ | | √ | | √ | | √ | | |
| PCA (Wang et al., 2019b) | | √ | √ | √ | √ | | √ | | √ | | |
| CIE (Yu et al., 2020) | | √ | √ | √ | √ | | √ | | √ | | |
| NGM (Wang et al., 2021a) | √ | | √ | √ | √ | √ | √ | | √ | | |
| LCS (Wang et al., 2020c) | √ | | √ | √ | √ | √ | √ | | √ | | |
| BBGM (Rolínek et al., 2020) | √ | | √ | √ | √ | | √ | | √ | | |
| GANN (Wang et al., 2020b) | | √ | √ | | | | √ | | | √ | |
| **RGM (ours)** | √ | | | | | √ | | √ | | | √ |

***Remark:** Despite the benefits of label-free and timely node correspondence generation, our RL module cannot be combined with front-end models for joint feature and solver learning, as mentioned in Sec. 1: under the RL framework, at least by our presented reward based on the affinity objective score, it is impossible to jointly train the front-end models with the back-end solver. This is because the reward itself is a function w.r.t. the model parameters of the front end appearance and structure model e.g. CNN/GNN. While in the supervised NGM (Wang et al., 2021a), the ground truth node correspondences as used for loss are computationally irrelevant to the front-end modules thus joint learning is feasible. Only NGM (Wang et al., 2021a)/LCS (Wang et al., 2020c) (these concurrent works are essentially similar at core idea) and RGM can be directly applied to QAP given an input affinity matrix. See results in experiments on QAPLIB in Sec. 5.3.*

## A  APPENDIX

### A.1  DISCUSSION WITH EXISTING WORK

Instead of the Lawler's QAP, several GM works choose the Koopmans-Beckmann's QAP (Koopmans & Beckmann, 1957), which requires the explicit input of two graphs. We argue that such raw information may not always be available in practice e.g. for privacy. For graph matching, the most general form is Lawler's QAP (Lawler, 1963) whose input is the pairwise affinity matrix whereby the raw graph information is removed and the Koopmans-Beckmann's QAP is a special case for Lawler's QAP. There are also standing and widely adopted public benchmarks for Lawler's QAP e.g. QAPLIB (Burkard et al., 1997). For its generality and popularity, SOTA deep GM works (Wang et al., 2020c; 2021a; Rolínek et al., 2020) follow this line, whereby a GNN model is applied on the so-called association graph whose weighted adjacency matrix is the affinity matrix. This GNN model is trained for node embedding on the association graph, which selects the node correspondence via supervised learning in one shot. Since Lawler's QAP only requires the affinity matrix and keeps the node and edge information of user unknown, the privacy can be better retained than explicitly using the input graph as done in the Koopmans-Beckmann's QAP. Therefore, Lawler's QAP has been adapted to several privacy-sensitive tasks, including model fusion (Liu et al., 2022), bioinformatics (Zaslavskiy et al., 2009), and text alignment (Fey et al., 2020b).

As mentioned in the paper, utilizing label-free revocable RL with affinity regularization for designing a new back-end solver becomes a promising tool for pushing the frontier of graph matching research. However, here we emphasize our method is focused on the back-end part whose input is the affinity matrix, and our method cannot be combined with learnable CNN and GNN for input graph feature extraction and metric learning part, for joint differentiable front-back-end learning. The reason is that, as will be shown in our approach, the RL reward is parameterized by the front-end CNN/GNN/MLP, which makes end-to-end impossible. For the above reason, our RL solver is trained on the input affinity matrix, by fixing the parameters of the front-end models which can be pretrained via existing supervised methods (Wang et al., 2019b). Table 5 compares existing works for their learning modules and techniques. This protocol is akin to the QAP learning part in (Wang et al., 2021a), which can also be regarded as the inherent limitation with RL. Fortunately, as will be shown in our experiments, our method still can outperform end-to-end supervised methods, especially with a large ratio of outliers. Our two-stage training pipeline is also more efficient than joint learning, and thus suitable for our method as RL is more costly than supervised learning.

---

**Algorithm 1:** Training algorithm for RGM. It also consists of revocable action (Sec. 4.2), inlier count information (Sec. 4.3.1), and affinity regularization (Sec. 4.3.2).

---

**Input:** Dataset $\mathbb{D}$; step size $\eta$; exploration rate $\epsilon$; updating frequency $c_1, c_2$; inlier count $n_i$; max round $T_{max}$; early stop round $T_{es}$; reward penalty $r_p$.

**Output:** Well trained Q-value network $f_\theta$.

1 **for** *episode* $\longleftarrow$ *0, 1, 2, . . .* **do**
2      Sample pairwise graphs $G^1, G^2$ from dataset $\mathbb{D}$;
3      Construct the association graph $G^a$ from $G^1 G^2$, and get its affinity matrix $\mathbf{K}$;
4      Acquire the initialization state $s$ of $G^a$;
5      **for** *ind* $\longleftarrow$ *0, 1, 2, . . . , $T_{max}$* **do**
6          *# Estimate Q-value:*
7          **if** *Affinity Regularization* **then**
8              Calc. $\mathbf{K}_{reg}$ for regularization by Eq. 7;
9              Input $\mathbf{K}_{reg}$ and $s$ to GNN in Eq. 9;
10          **else**
11              Input $\mathbf{K}$ and $s$ to GNN in Eq. 9;
12          Get $\mathbf{Q}$ from the Q-value network in Eq. 10;
13          ——————————————————————-
14          *# Choosing next action:*
15          **if** *Revocable Mechanism* **then**
16              Set available vertices set $\mathbb{V}$ to the whole vertices $V^a$ of the association graph $G^a$;
17          **else**
18              Calculate available vertices set $\mathbb{V}$ by Eq. 3;
19          With probability $\epsilon$ select a random action $a \in \mathbb{V}$ otherwise select $a = \arg\max_{a \in \mathbb{V}} \mathbf{Q}(s, a; f_\theta)$;
20          ——————————————————————-
21          *# Interacting with the environment:*
22          $s' \overset{a}{\longleftarrow} s$;
23          **if** *Affinity Regularization* **then**
24              $r = J(s') \cdot f(|s'|) - J(s) \cdot f(|s|)$;
25          **else**
26              $r = J(s') - J(s)$;
27          **if** *Revocable Mechanism* **then**
28              $r = r - r_p$;
29          Store the transition $(s, a, r, s')$ in $\mathcal{M}$;
30          ——————————————————————-
31          *# Updating the neural networks:*
32          $cnt \leftarrow cnt + 1$;
33          **if** $cnt \,\% \, c_1 == 0$ **then**
34              Calculate $\mathcal{L}(f_\theta; \mathcal{M})$ by Eq. 12;
35              Update $f_\theta : \theta \leftarrow \theta - \eta \nabla_\theta \mathcal{L}(f_\theta; \mathcal{M})$;
36          Update the transition priority in $\mathcal{M}$;
37          **if** $cnt \,\% \, c_2 == 0$ **then**
38              Update $f_{\theta^-} : f_{\theta^-} \leftarrow f_\theta$;
39          ——————————————————————-
40          *# Transition:*
41          $s \leftarrow s'$;
42          **if** *(Inlier Count and $|s| == n_i$) or (current best solution unchanged in $T_{es}$ rounds)* **then**
43              **break;**

## A.2 Details of the proposed method

The idea of RL is to learn from the interactions between the agent and the environment (Sutton & Barto, 2005). The agent's observation of the current environment is called state $s$. The agent chooses an action $a$ given the current state by a specific policy. After the agent performs an action, the environment will transfer to another state $s'$. Meanwhile, the environment will feedback to the agent with a reward $r$. This pipeline solves the problem progressively. For graph matching, "progressively" means to select the vertex in the association graph one by one. The environment is defined as a partial solution to the original combinatorial problem (Eq. 1) and equivalently the association graph, where the reward denotes the improvement of the objective function by matching a new pair of nodes. The interactions between the agent and the environment are recorded as a transition $(s, a, r, s')$ into an experience replay memory $\mathcal{M}$. After several episodes, the agent updates its networks $f_\theta$ according to the transitions sampled from $\mathcal{M}$.

### A.2.1 Network Structure

**1) State Representation Networks.** To better represent the current state on the association graph, we choose graph neural networks (GNN) (Kipf & Welling, 2017) to compute its embedding. GNN extracts the vertex features based on their adjacent neighbors. To better use the edge weights in the association graph, we derive from the idea of struct2vec (Dai et al., 2016). In our embedding networks, the current solution, node weights, and edge weights of the association graph are considered. The embedding formula is:

$$
\begin{aligned}
\mathbf{E}^{t+1} &= \text{ReLU}(\mathbf{h}_1 + \mathbf{h}_2 + \mathbf{h}_3 + \mathbf{h}_4) \\
\mathbf{h}_1 &= \mathbf{X}' \cdot \theta_1^\top, \quad \mathbf{h}_2 = \frac{\mathbf{A} \cdot \mathbf{E}^t}{(n_1 - 1)(n_2 - 1)} \cdot \theta_2 \\
\mathbf{h}_3 &= \frac{\mathbf{A} \cdot \mathbf{F} \cdot \theta_3^\top}{(n_1 - 1)(n_2 - 1)}, \quad \mathbf{h}_4 = \frac{\sum \text{ReLU}(\mathbf{W} \cdot \theta_5)}{(n_1 - 1)(n_2 - 1)} \cdot \theta_4
\end{aligned}
\tag{9}
$$

where $\mathbf{E}^t \in \mathbb{R}^{n_1 n_2 \times d}$ denotes the embedding in the $t$-th iteration, with $d$ as the hidden size. At every iteration, the embedding is calculated by four hidden parts $\mathbf{h}_1, \mathbf{h}_2, \mathbf{h}_3, \mathbf{h}_4 \in \mathbb{R}^{n_1 n_2 \times d}$. $\theta_1 \in \mathbb{R}^d$, $\theta_2 \in \mathbb{R}^{d \times d}$, $\theta_3 \in \mathbb{R}^d$, $\theta_4 \in \mathbb{R}^{d \times d}$ and $\theta_5 \in \mathbb{R}^d$ are the weight matrices in the neural networks. $t$ is the index of the iteration and the total number of the iterations is $T$. We set the initial embedding $\mathbf{E}^0 = \mathbf{0}$ and use ReLU as the activation function.

As for the hidden parts, each hidden part represents a kind of feature: $\mathbf{h}_1$ is to calculate the impact of current permutation matrix $\mathbf{X}'$ which is transformed from the current partial solution $\mathbb{U}'$. $\mathbf{h}_2$ is to take neighbor's embedding into consideration, where $\mathbf{A}$ is the adjacency matrix of the association graph and divide $(n_1 - 1)(n_2 - 1)$ is for average since every vertex has $(n_1 - 1)(n_2 - 1)$ neighbors. $\mathbf{h}_3$ calculates the average of neighbor's vertex weights, where $\mathbf{F}$ is the vertex weight matrix. $\mathbf{h}_4$ is designed to extract the features of adjacent edges, where $\mathbf{W}$ is the edge weight matrix. Please note that the core inputs of our GNN are the permutation matrix $\mathbf{X}$ and the affinity matrix $\mathbf{K}$ ($\mathbf{A}$, $\mathbf{F}$ and $\mathbf{W}$ are derived from the affinity matrix $\mathbf{K}$).

**2) Q-Value Estimation Networks.** The Q-learning based algorithms use $\mathbf{Q}(s, a)$ to represent the value of taking action $a$ in state $s$, as an expected value of the acquired reward after choosing this action. The agent picks the next action given the estimation of $\mathbf{Q}(s, a)$. The Q-value estimation network $f_\theta$ takes the embedding of the current state as input and predicts the Q-value for each possible action. We adopt Dueling DQN (Wang et al., 2016) as our approximator to estimate the Q-value function. The architecture of our $f_\theta$ is:

$$
\begin{aligned}
\mathbf{h}_5 &= \text{ReLU}(\mathbf{E}^\top \cdot \theta_6 + b_1), \quad \mathbf{h}_v = \frac{\sum \mathbf{h}_5 \cdot \theta_7}{n_1 n_2} + b_2, \\
\mathbf{h}_a &= \mathbf{h}_5 \cdot \theta_8 + b_3, \quad \mathbf{Q} = \mathbf{h}_v + \left( \mathbf{h}_a - \frac{\sum \mathbf{h}_a}{n_1 n_2} \right)
\end{aligned}
\tag{10}
$$

where $\mathbf{E}^\top$ is the final output of the embedding network by Eq. 9. $\mathbf{h}_5 \in \mathbb{R}^{n_1 n_2 \times d}$ is the hidden layer for embedding. $\mathbf{h}_v \in \mathbb{R}^1$ is the hidden layer for the state function. $\mathbf{h}_a \in \mathbb{R}^{n_1 n_2}$ is the hidden layer for the advantage function. $\theta_6 \in \mathbb{R}^{d \times d}, \theta_7 \in \mathbb{R}^d, \theta_8 \in \mathbb{R}^d$ are the weights of the neural networks. $b_1$, $b_2$, and $b_3$ are the bias vectors. $\mathbf{Q} \in \mathbb{R}^{n_1 n_2}$ is the final output of our Q-value estimate network. It predicts the value of each action given the current state.

The state function and advantage function are designed to separate the value of state and action. Specifically, the state function predicts the value of different states and the advantage predicts the value of each action given the particular state. The previous work (Wang et al., 2016) shows that this dueling architecture can better learn the impact of different actions. Besides, we force the sum of the output vector of the advantage function to 0 by subtracting the mean of it, which makes the separation of the state value and advantage easier. We use $\mathbf{Q}(s, a; f_\theta)$ to denote the estimated Q-value by $f_\theta$ when the agent takes action $a$ on state $s$.

### A.2.2 Experience Replay Memory

For sample efficiency, we maintain a prioritized experience replay memory $\mathcal{M}$ (Schaul et al., 2016) that stores the experience of the agent, defined as the transition $(s_i, a_i, r_i, s'_i)$ (denoting state, action, reward, and state of next step respectively). As the training progresses, we add new transitions to $\mathcal{M}$ and remove old transitions. The agents will take samples from the experience replay memory to update their neural networks. We follow the idea of prioritized experience replay memory, which adds a priority for each transition and higher priority denotes higher probabilities to be sampled:

$$P(i) = \frac{(p_i)^\alpha}{\sum_j (p_j)^\alpha} \tag{11}$$

where $P(i)$ is the probability and $p_i$ is the priority of the $i$-th transition. $\alpha$ is a hyperparameter. The calculation of $p_i$ is based on the underfitting extent of the transition, and a larger bias rate of the agent's Q-value estimation means a relatively higher priority.

### A.2.3 Model Updating

We follow Double DQN (Hasselt et al., 2016) to calculate the loss function and update the parameters. We pick the next action $a'$ by the current Q-value estimate network $f_\theta$, but use the target Q-value estimate network $f_{\theta^-}$ to predict its value as Eq. 12 shows. The motivation of designing this loss function is: the Q-value that is overestimated in one network will be mitigated to an extent in another network.

$$a' = \arg \max_{a'} \mathbf{Q}(s', a'; f_\theta) \tag{12}$$

$$\mathcal{L}(f_\theta; \mathcal{M}) = \mathbb{E}_{s,a,r,s' \sim \mathcal{M}} \left[ \left( r + \gamma \mathbf{Q}(s', a'; f_{\theta^-}) - \mathbf{Q}(s, a; f_\theta) \right)^2 \right]$$

where $\mathcal{L}(\cdot)$ is the loss function to be optimized. $f_\theta$ is the current Q-value estimation network and $f_{\theta^-}$ is the target Q-value estimation network. $s, a, r, s', a'$ stand for the state, action, reward, next state, and next action. The target network $f_{\theta^-}$ shares the same architecture with $f_\theta$ and the parameters of $f_{\theta^-}$ will be replaced by the parameters of $f_\theta$ every period. The design of such a target network is for keeping the target Q value remains unchanged for a period of time, which reduces the correlation between the current Q value and the target one and improves the training stability.

### A.2.4 Further Discussion of the Revocable Action Framework

Note that our revocable action mechanism requires most changes in our environment settings. Therefore, we design a new RL environment for the revocable action mechanism, and the main differences from the original environment are: the available set is gone; the agent can choose any vertex in the vertices $V^a$ of the association graph $G^a$; when the environment receives a vertex that is in conflict with the current partial solution, it releases the conflicted vertices and adds the new-coming vertex to the partial solution. The agent design is irrelevant to the basic or the revocable environment, and the training process of the revocable framework almost remains the same, as Alg. 1 shows. While the revocable flexibility also brings some side effects, e.g. we can not adopt the acceleration tricks for GNN, such as dynamic embedding (Wang et al., 2021b), which can otherwise speedup the inference time.

As for the stopping rules for our revocable action mechanism, there are three stopping cases and the current best solution will be returned: 1) when the number of matched nodes equals to the given or estimated input inlier count; 2) no affinity score improvement is made (the current best solution remains the same) in $T_{es}$ rounds where $T_{es}$ is the given hyperparameter for early stopping; 3) the number of rounds reaches the preset max round parameter $T_{max}$. Besides, we add a small reward penalty $r_p$ in each step to make the agent away from repeatedly taking an action and canceling it. The usage of the these parameters can be seen in Alg. 1.

To our best knowledge, there are in general two existing techniques allowing revocable actions, at least for RL-based combinatorial optimization: Local Rewrite (Chen & Tian, 2019) and ECO-DQN (Barrett et al., 2020). Here we discuss our difference from these methods.

The local rewrite framework keeps improving the solution given as input by exchange the parts of it. However, the performance of the local rewrite highly relies on the input solution. In our empirical tries, the efficiency and effectiveness of local rewrite are unsatisfactory, as will be shown in our experiments. The ECO-DQN framework does have a promising performance in the Maximum Cut problem, but it is inherently designed to work on this specific Maximum Cut problem, which only has fewer or no constraints, and is clearly impossible to adapt to the graph matching problem. Therefore, in this paper, we devise a new revocable framework RGM to meet the characteristic of the graph matching problem, which is more friendly to graph with relatively more constraints. To verify the effectiveness of our proposed revocable framework, we compare it with the local rewrite framework in our experiments in Sec. A.4.6.

We believe our devised scheme for revocable action is suited to the setting when the graph size is moderate to afford such a costly scheme, while the constraints are relatively heavy and complex to make the revocable action necessary, whereby graph matching has become a suited problem setting.

## A.3 Experiments protocol

We perform experiments on various benchmarks including image data as well as pure combinatorial optimization problem instances, as the latter is especially suited for our back-end solver. We evaluate the robustness against outliers, as the (ground truth or estimated) number of inliers is given as hyperparameter. Whenever this information is known or not, one can always apply our proposed regularization technique to improve its robustness against outlier. The experiments are conducted on a Linux workstation with NVIDIA 2080Ti GPU and AMD Ryzen Threadripper 3970X 32-Core CPU with 128G RAM. Note that the graph size in our experiments is mostly more than 11 and the enumeration of all permutations is at the scale of more than 20 million which means an exhaustive search is impossible on commodity computers.

### A.3.1 Hyperparameter Settings

For the hyper parameters, we set $\gamma = 0.9$ in Eq. 12, the target network update frequency as 40, the replay size as 100,000. For the greedy part, the greedy rate $\epsilon$ decays from 1.0 to 0.02 in 20,000 episodes. For the learning module, we set 1e-5 as the learning rate and 64 as the batch size. The hidden size of our GNN is 128 and the number of layers $T$ is 3. The hidden size of the Q value network is 64. For the affinity regularization module, the range of the data points used for approximation is $S = [n_x - 2, n_x + 2]$.

### A.3.2 Evaluation Metrics

For testing, given the affinity matrix $\mathbf{K}$, RGM predicts a permutation matrix $\mathbf{X}^{pred} \in \{0, 1\}^{n_1 \times n_2}$ transformed from its solution set $\mathbb{U}$. Based on $\mathbf{X}^{pred}$ and ground truth $\mathbf{X}^{gt} \in \{0, 1\}^{n_1 \times n_2}$ (note that $\sum \mathbf{X}^{gt}$ equals the number of inliers, since the rows and columns of outliers are always zeros.). Two evaluation metrics are used: objective affinity score, and F1 score:

$$\textbf{Objective score} = \frac{\text{vec}(\mathbf{X}^{pred})^\top \mathbf{K} \text{ vec}(\mathbf{X}^{pred})}{\text{vec}(\mathbf{X}^{gt})^\top \mathbf{K} \text{ vec}(\mathbf{X}^{gt})} \tag{13}$$

$$\text{Recall} = \frac{\sum \left( \mathbf{X}^{pred} * \mathbf{X}^{gt} \right)}{\sum \mathbf{X}^{gt}} \tag{14}$$

$$\text{Precision} = \frac{\sum \left( \mathbf{X}^{pred} * \mathbf{X}^{gt} \right)}{\sum \mathbf{X}^{pred}} \tag{15}$$

$$\textbf{F1 score} = \frac{2 \cdot \text{Recall} \cdot \text{Precision}}{\text{Recall} + \text{Precision}} \tag{16}$$

where $*$ denotes element-wise matrix multiplication. Note that the defined objective score here is agnostic to the presence of outliers, which is a common protocol used in existing works. Specifically, for a traditional affinity matrix as used in previous works, its elements are set non-negative, and thus the solver generally aims to match node correspondences as many as possible.

We also conduct experiments on the well-known QAPLIB dataset. For the problem instances on QAPLIB, the goal is to minimize the objective score. Besides, the ground truth solution is supposed

unknown due to its NP-hard nature. Therefore, we use the gap between the score of the predicted solution and the optimal score provided in the benchmark which is continuously updated by uploaded new best solutions, as the metric:

$$\textbf{Optimal gap} = \frac{\text{vec}(\mathbf{X}^{pred})^\top \mathbf{K} \, \text{vec}(\mathbf{X}^{pred}) - \text{optimal}}{\text{optimal}} \tag{17}$$

Note that in the QAP test, there is no outlier issue, and our solver purely optimizes the objective score, by matching all the nodes.

### A.3.3 COMPARED METHODS

As mentioned before, RGM falls in line with learning-free graph matching back-end solvers that use the affinity matrix $\mathbf{K}$ as input, regardless $\mathbf{K}$ is obtained by learning-based methods or not. Both traditional methods and learning-based methods are compared:

**GAGM** (Gold et al., 1996) utilizes the graduated assignment technique with an annealing scheme, which can iteratively approximate the cost function by Taylor expansion.

**RRWM** (Cho et al., 2010) proposes a random walk view of the graph matching, with a re-weighted jump on graph matching.

**IFPF** (Leordeanu et al., 2009) iteratively improves the solution via integer projection, given a continuous or discrete solution.

**PSM** (Egozi et al., 2013) improves the spectral matching by presenting a probabilistic interpretation of the spectral relaxation scheme.

**GNCCP** (Liu et al., 2012) follows the convex-concave path-following scheme, with a simpler form of the partial permutation matrix.

**BPF** (Wang et al., 2018) designs a branch switching technique to seek better paths at the singular points, to deal with the singular point issue in the previous path following strategy.

**ZACR** (Wang et al., 2020a) designs to suppress the matching of outliers by assigning zero-valued vectors to the potential outliers, which is the latest graph matching solver designated for outliers.

In particular, we further compare RGM with current popular deep graph matching methods: **GMN** (Zanfir et al., 2018), **PCA** (Wang et al., 2019b), **NGM** (Wang et al., 2021a), **LCS** (Wang et al., 2020c), **BBGM** (Rolínek et al., 2020), which are the state-of-the-art deep graph matching methods, and more importantly, most of them are all open-sourced which are more convenient for a fair comparison.

### A.3.4 DATASETS FOR EVALUATION

We briefly describe the used datasets, in line with the recent comprehensive evaluation for deep GM (Wang et al., 2021a).

**Synthetic Dataset** is created by random 2D coordinates as nodes and their distances as edge features for graph matching. Specifically, one graph is randomly constructed to which random noise is further added to generate the other graph for matching. The ground truth matching normally is set as the identity matrix.

**Willow Object** is collected from real images by Cho et al. (2013). It contains 256 images from 5 categories, and each category is represented with at least 40 images. All instances in the same class share 10 distinctive image keypoints whose correspondences are manually labeled as ground truth. For testing the performance of handling outliers, we add several random outliers to each image.

**Pascal VOC** (Bourdev & Malik, 2009) consists of 20 classes with keypoint labels on natural images. The instances vary by scale, pose and illumination. The number of keypoints in each image ranges from 6 to 23.

**QAPLIB** (Burkard et al., 1997) contains 134 real-world QAP instances from 15 categories, e.g. planning a hospital facility layout or testing of self-testable sequential circuits. The problem size is defined as $n_1 = n_2$ by Lawler's QAP. We use 14 of the 15 categories, the only one left is "els", as there is only one sample in this category.

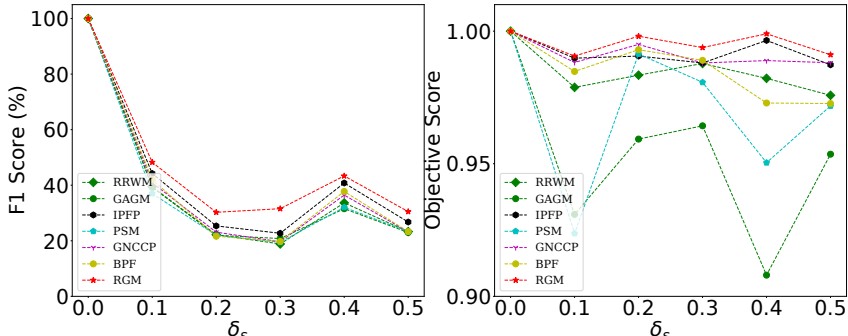

Figure 5: Performance comparison w.r.t F1 score and objective score (the higher the better) by increasing noise level on the synthetic dataset.

### A.3.5 TRAINING AND TESTING PROTOCOLS

Due to the complexity in evaluation of learning-based graph matching methods and QAP solvers, here we elaborate the training and testing protocols in detail. We use the open-source version of the compared methods, and we tune or follow the hyperparameters set by the authors, to achieve the sound performance.

For the synthetic dataset, we use geometry features to construct the affinity matrix with the train-test split rate (2 : 1) follows Jiang et al. (2021).

For the natural image dataset Willow Object and Pascal VOC, we use the pretrained features by CNN and GNN from BBGM (Rolínek et al., 2020) via supervised learning on the training set, and use the pre-splitted train/test set (8 : 1) in line with BBGM as well. We input the learned affinity matrix to RGM and all learning-free methods[2], to make the comparison with supervised methods as fair as possible.

For QAPLIB, there is no need for front-end feature extractors as the affinity matrix is already given. The train-test split rate for RGM is (1 : 1) for each of the selected 14 categories, as we choose the smaller-size half of the instances to train RGM in that category. While for the peer method NGM (Wang et al., 2021a), due to its model's nature, it does not split the train-test set in their QAPLIB experiments and test directly after training on the same set. In contrast, our RGM follows the basic protocol in RL, which splits the train-test set to make a relatively fair comparison with the baselines.

### A.4 ADDITIONAL EXPERIMENTS

### A.4.1 EXPERIMENTS ON SYNTHETIC DATASET

We evaluate RGM on the synthetic graphs following the protocol of Wang et al. (2021a). The synthetic data test is relatively simple, and it is mainly to show the effectiveness of our back-end solver when the front-end information is limited as there is no visual data for CNN to learn. More outlier tests will be given on the real data.

We first generate sets of random points in the 2D plane. The coordinates of these points are sample from uniform distribution $U(0, 1) \times U(0, 1)$. First, we select 10 sets of points as the set of ground truth points. Then, we randomly scale their coordinates from $U(1 - \delta_s, 1 + \delta_s)$. The set of scaled points and the set of ground truth points are regarded as the pairwise graphs to be matched. We set there are 10 inliners without outlier, and the noise level $\delta_s$ varies from 0 to 0.5. For the calculation of affinity matrix $\mathbf{K}$, the node affinity is set by 0 and the edge affinity is set by the difference of edge length: $\mathbf{K}_{ia,jb} = \exp(-\frac{(f_{ij} - f_{ab})^2}{\sigma_1})$, where the $f_{ij}$ is the edge length of $E_{ij}$. We generate 300 sets of scaled points for each ground truth sets and get 3,000 pairwise graphs. We split the data into the training and testing sets by the ratio of (2 : 1).

The results of the synthetic datasets are shown in Fig. 5. Evaluations are performed in terms of the noise level $\delta_s$. We can see that RGM performs the best in terms of matching F1 score and objective score in all experiments.

---

[2]ZACR is an exception which will be explained in Sec. A.4.2 in detail.

Table 6: Performance comparison w.r.t F1 score and objective score (the higher the better) in the Willow Object dataset, where "F1" and "Obj" are short for F1 score and objective score. All images contain 10 inliers and **3** randomly generated outliers in both graphs. For our RGM, "RGM + AR" means RGM with affinity regularization, "RGM + IC" means RGM with inlier count information, and "RGM + AR + IC" means RGM with both. We add the ablation study about RGM without the revocable mechanism in the last three column of the table, denoted by "w/o rev.".

| | Class | Car | | Duck | | Face | | Motorbike | | Winebottle | | **Average** | |
|---|---|---|---|---|---|---|---|---|---|---|---|---|---|
| | Method | F1 | Obj | F1 | Obj | F1 | Obj | F1 | Obj | F1 | Obj | F1 | Obj |
| Learning-free | RRWM (Cho et al., 2010) | 65.48% | 0.8039 | 60.25% | 0.7165 | 84.21% | 0.9665 | 54.04% | 0.7332 | 66.19% | 0.7810 | 66.03% | 0.7989 |
| | GAGM (Gold et al., 1996) | 57.50% | 0.7186 | 53.59% | 0.6778 | 82.87% | 0.9478 | 47.38% | 0.6325 | 60.96% | 0.7085 | 60.46% | 0.7365 |
| | IPFP (Leordeanu et al., 2009) | 71.16% | 0.8294 | 62.29% | 0.7233 | 84.65% | 0.9695 | 60.07% | 0.7344 | 68.86% | 0.7905 | 69.41% | 0.8063 |
| | PSM (Egozi et al., 2013) | 66.64% | 0.8047 | 61.22% | 0.7343 | 82.79% | 0.9336 | 55.63% | 0.7103 | 66.02% | 0.7831 | 66.46% | 0.7914 |
| | GNCCP (Liu et al., 2012) | 73.65% | 0.8484 | 63.71% | 0.7459 | 84.38% | 0.9628 | 62.20% | 0.7314 | 71.43% | 0.7966 | 71.07% | 0.8164 |
| | BPF (Wang et al., 2018) | 74.00% | 0.8465 | 62.91% | 0.7342 | 84.38% | 0.9628 | 63.09% | 0.7354 | 71.90% | 0.7880 | 71.26% | 0.8131 |
| | ZACR (Wang et al., 2020a) | 66.27% | 0.7899 | 61.67% | 0.6891 | 84.59% | 0.9531 | 59.27% | 0.6881 | 68.10% | 0.7937 | 67.98% | 0.7824 |
| Supervised | GMN (Zanfir et al., 2018) | 57.96% | - | 57.87% | - | 86.66% | - | 58.18% | - | 67.52% | - | 65.64% | - |
| | PCA (Wang et al., 2019b) | 71.00% | - | 57.80% | - | 86.12% | - | 57.89% | - | 65.74% | - | 67.71% | - |
| | LCS (Wang et al., 2020c) | 72.23% | - | 61.90% | - | 86.84% | - | 62.35% | - | 69.15% | - | 70.49% | - |
| | NGM (Wang et al., 2021a) | 68.77% | - | 61.09% | - | 86.58% | - | 55.65% | - | 69.48% | - | 68.31% | - |
| | BBGM (Rolínek et al., 2020) | 76.10% | - | 63.62% | - | 96.71% | - | 60.93% | - | 69.70% | - | 73.41% | - |
| | NGM-v2 (Wang et al., 2021a) | 78.76% | - | 65.41% | - | 86.84% | - | 63.94% | - | 71.19% | - | 73.23% | - |
| RL | RGM + AR | 78.14% | **0.9543** | 63.46% | **0.8932** | 97.06% | **0.9871** | 65.88% | **0.8858** | 73.73% | **0.9236** | 75.70% | **0.9288** |
| | RGM + IC | 80.60% | 0.8809 | 63.80% | 0.8369 | 97.30% | 0.9838 | 68.60% | 0.8603 | 73.30% | 0.8683 | 76.70% | 0.8860 |
| | RGM + AR + IC | **81.65%** | 0.9389 | **66.57%** | 0.8685 | **97.80%** | 0.9838 | **67.30%** | 0.8639 | **74.53%** | 0.9061 | **77.57%** | 0.9122 |
| | RGM + AR w/o rev. | 71.20% | 0.8966 | 56.48% | 0.8302 | 96.32% | 0.9342 | 66.90% | 0.8250 | 70.04% | 0.8649 | 72.19% | 0.8702 |
| | RGM + IC w/o rev. | 70.10% | 0.8309 | 56.50% | 0.7869 | 94.73% | 0.9338 | 69.18% | 0.8103 | 71.57% | 0.8183 | 72.42% | 0.8360 |
| | RGM + AR + IC w/o rev. | 72.69% | 0.8880 | 59.58% | 0.8181 | 96.91% | 0.9338 | 69.11% | 0.8139 | 71.30% | 0.8552 | 73.92% | 0.8618 |

### A.4.2 PERFORMANCE ON THE WILLOW OBJECT DATASET

Table 6 shows the results over the five categories give three outliers. We compare our methods (bottom box) with the learning-free back-end solvers (top box) and the learning based deep graph matching methods (middle box). We use learning features extracted by BBGM (Rolínek et al., 2020) to construct the affinity matrix, which is used as input for all learning-free back-end solvers and our methods. For other learning-based baselines, we train and test them by their own pipelines directly, and they do not report their objective score because learning-based methods only care about the accuracy or F1 score. Since there are several outliers that should not be matched, the ground truth matching results only contain parts of the input keypoints (10 actually). Therefore, we use the F1 score instead of accuracy to test the performance.

One may attribute the advantage of "RGM + IC" and "RGM + AR + IC" to the use extra information of the inlier count which is a unique ability of our RL-based model compared to peer baselines. Yet "RGM + AR" does not require the inlier count information as input, and can still outperform all baselines in almost all settings.

We find a strange result that BPF (Wang et al., 2018) reaches the best performance rather than ZACR (Wang et al., 2020a), which is the latest GM solvers tailored for handling outliers. Per discussion with the authors of ZACR, this is perhaps mainly due to a few strong assumptions they made which is more suitable to the 50 images (30 cars and 20 motorbikes) they used for experiments in their paper, which may not always hold in other datasets including Willow Object. For example, ZACR requires edges linked by two inliers have clear higher similarities than the edges linked by inlier-outlier or outlier-outlier. Besides, by its inherent design, ZACR solves Koopmans-Beckmann's QAP instead of Lawler's QAP, and therefore has some difficulty utilizing the affinity matrix which is obtained by pretraining BBGM. Via the communication and discussion with the authors of ZACR, we have tried to modify their code including using the learned node and edge features inline with their model's intereface. Regrettably, the results in Table 6 and Table 1 are the best results we can get.

The matching visualization is given in Fig. 6. We visualize the matching results of RGM on all five categories. We paint the inliers as green nodes and the outliers as blue nodes. In each pair of images, the green and red lines represent correct and incorrect predictions respectively. Since it is supposed to match all inliers and ignore the outliers, we can see that RGM barely matches the blue outliers and focuses on the green inliers.

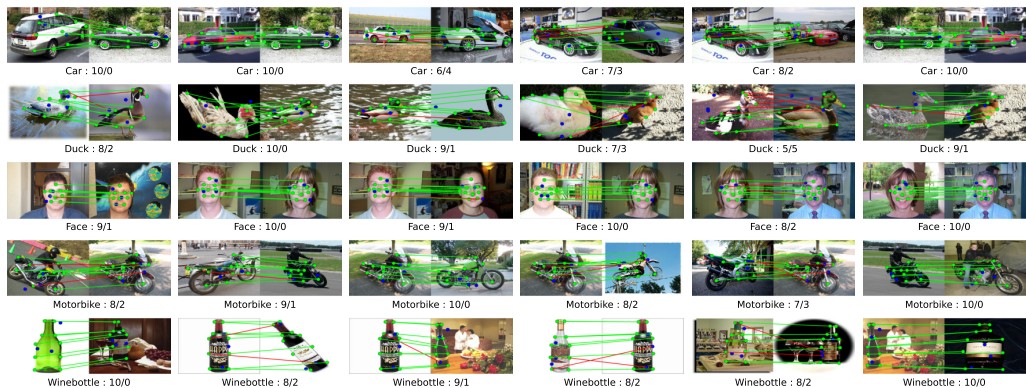

Figure 6: Visual illustration of the matching results by RGM on the Willow Object dataset with 10 inliers (green), and 3 outliers (blue) which are randomly extracted from the images. Green and red lines represent correct and incorrect node matchings respectively. The correct solution is supposed to match all green inliers with green line. The subtitle of each figure shows the correct / incorrect matching count out of the 10 inliers.

| | | **Testing Amount of Outliers** | | | | | |
|---|---|---|---|---|---|---|---|
| | **0** | **1** | **2** | **3** | **4** | **5** | **6** |
| **0** | 90.8% | 87.1% | 78.5% | 76.1% | 68.1% | 63.5% | 60.9% |
| **1** | 89.9% | 87.7% | 79.1% | 75.9% | 69.8% | 65.1% | 62.0% |
| **2** | 89.5% | 86.8% | 80.4% | 77.3% | 71.2% | 64.5% | 64.5% |
| **3** | 88.1% | 87.2% | 79.8% | 77.6% | 72.6% | 67.1% | 63.9% |
| **4** | 87.9% | 86.5% | 79.3% | 76.5% | 73.8% | 66.9% | 64.3% |
| **5** | 86.4% | 85.7% | 78.5% | 77.1% | 72.9% | 67.5% | 64.7% |
| **6** | 85.0% | 85.6% | 77.9% | 76.3% | 72.3% | 66.1% | 65.1% |

*Training Amount of Outliers*

Figure 7: Generalization test for number of outliers by F1 Score (↑). Row and column indices denote the amount of outliers on training and testing set, respectively. The average F1 on all five classes in Willow Object is reported. For each testing set (column), the darker red the better.

### A.4.3 GENERALIZATION

**1) Generalization to different amounts of outliers.** We carry additional experiments to test the generalization ability among different numbers of outliers. In this study, we train our RGM on one certain number of outliers and test it on another setting. We conduct these experiments on Willow Object with 10 inliers and a range of outliers from 0 to 6. The results are shown in Fig. 7, where for every testing case (column) darker red means better performance. We can see that our RGM can generalize well to the different numbers of outliers, since the performance of RGM is promising. Relatively speaking, training RGM with 2, 3, or 4 outliers can reach a better generalization performance, and training RGM with 3 outliers can reach the best.

**2) Generalization among similar categories.** Fig. 8 (a) shows the generalization ability of RGM among similar categories. We use one class for training and another class for testing. For every testing class (every column) the red is the darker the better. We can see RGM generalizes well to different classes.

**2) Generalization on QAPLIB.** Fig. 8(b) shows the generalization ability of RGM, which is trained on one class and tested on another. For every testing class (every column), the darker red means the low optimal gap, the better performance. It shows that RGM generalizes soundly to unseen instances with different problem sizes.

### A.4.4 HYPERPARAMETER SENSITIVITY STUDY

We conduct the study on Willow Object with the same setting as aforementioned experiments, where there are 10 inliers and 3 outliers in each image. We choose six hyperparameters: batch size, $\gamma$, experience replay size, hidden size in GNN, hidden size in Q value network, and set the regularization function for affinity regularization by three function forms. The results in Fig. 9 show that RGM is not sensitive to hyperparameters in batch size, hidden size, and regularization function, since there

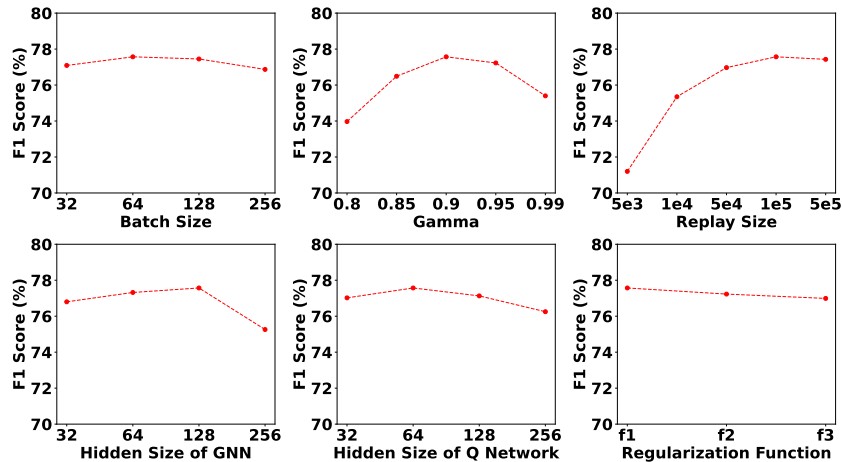

(a) Pascal VOC dataset

| Training Instance \ Testing Instance | Car | Bus | Chair | Sofa | Dog | Cat | Train | Tv |
|---|---|---|---|---|---|---|---|---|
| Car | 68.3% | 63.2% | 25.2% | 37.8% | 38.5% | 46.3% | 60.9% | 47.6% |
| Chair | 33.6% | 49.0% | 35.2% | 49.1% | 37.2% | 44.9% | 57.4% | 36.9% |
| Dog | 47.8% | 56.1% | 28.3% | 45.3% | 55.2% | 54.9% | 61.0% | 71.4% |
| Train | 51.0% | 68.0% | 30.3% | 44.1% | 41.3% | 52.9% | 88.7% | 83.0% |

(b) QAPLIB dataset

| Training Instance \ Testing Instance | bur(26) | chr(12-25) | esc(16-64) | had(12-20) | nug(12-30) | rou(12-20) | src(12-20) | tai(12-64) |
|---|---|---|---|---|---|---|---|---|
| bur | 7.1% | 129.1% | 48.5% | 8.3% | 10.3% | 22.9% | 56.7% | 19.0% |
| chr | 12.1% | 122.3% | 46.8% | 8.3% | 12.2% | 20.6% | 60.4% | 24.3% |
| esc | 15.5% | 133.2% | 44.2% | 8.0% | 15.3% | 21.1% | 60.4% | 27.3% |
| had | 16.9% | 124.2% | 65.7% | 7.0% | 11.4% | 23.3% | 51.3% | 20.3% |
| nug | 9.8% | 135.6% | 80.4% | 9.5% | 9.7% | 19.5% | 48.1% | 20.1% |
| rou | 11.4% | 130.4% | 51.3% | 8.9% | 14.7% | 16.4% | 56.9% | 27.4% |
| src | 16.4% | 122.4% | 51.4% | 10.1% | 12.3% | 21.2% | 44.9% | 25.3% |
| tai | 15.8% | 131.6% | 45.0% | 8.9% | 15.2% | 17.5% | 64.9% | 17.6% |

Figure 8: Generalization test w.r.t (a) F1 score (↑), (b) optimal gap (↓). Row and column indices denote training and testing classes, respectively. For every testing class (column), the darker red the better.

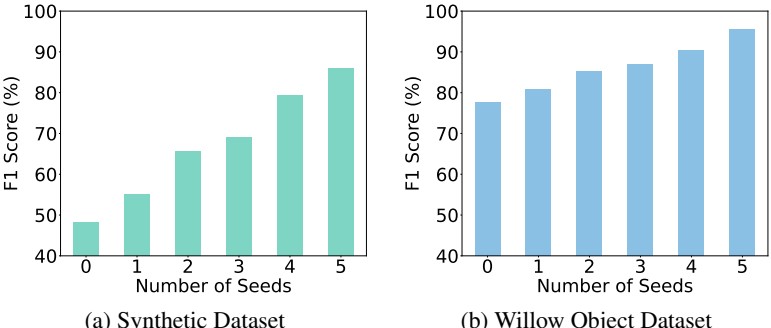

Figure 9: Hyperparameter sensitivity study for our revocable deep reinforcement learning: the results over different values of hyperparameters. The experiments are conducted on the Willow Object with 10 inliers and 3 outliers. The average F1 score (↑) on all five classes is reported.

(a) Synthetic Dataset  (b) Willow Object Dataset

Figure 10: Case study: performance comparison w.r.t F1 score (↑) in terms of different numbers of initial matched seeds (i.e. node pairs).

are only small fluctuations in F1 score. Here $\gamma$ is the hyperparameter in Eq. 12 as the rate for considering current reward and long-term reward, whose value cannot be too high or too low. RGM performs badly when the replay size is too small for experience replay.

### A.4.5 CASE STUDY: SEEDED GRAPH MATCHING

As aforementioned, the core of RGM is to learn the back-end decision making process for GM. Moreover, RGM can work flexibly by utilizing additional information e.g. the initial seeds, which mean that one or several nodes in each of the original pairwise graphs is already matched by human or other information sources. In implementation, we only need to set the initial seeds as the first several actions, and then let RGM execute normally.

We conduct this case study on both the synthetic dataset and the Willow Object dataset, of which each image contains 10 inliers and 3 outliers. Fig. 10 shows that adding suitable initial seeds does

Table 7: Comparison with the local rewrite (LR) (Chen & Tian, 2019) boosting technique on QAPLIB dataset w.r.t the optimal gap (the lower the better) and inference time in seconds, where "+LR" denotes using local rewrite given the output from the original methods, and "RGM w/o rev." denotes our method without the revocable mechanism as described in Sec. 4.2.

| | Time per | QAPLIB | | |
|---|---|---|---|---|
| | instance | mean ↓ | min ↓ | max ↓ |
| SM (Leordeanu & Hebert, 2005) | **0.01** | 181.22% | 46.93% | 949.94% |
| + LR | 47.51 | 119.14% | 44.21% | 291.15% |
| RRWM (Cho et al., 2010) | 0.15 | 169.50% | 49.51% | 432.94% |
| + LR | 47.93 | 100.19% | 46.25% | 223.84% |
| SK-JA (Kushinsky et al., 2019) | 563.44 | 93.23% | **9.03%** | 497.91% |
| + LR | 623.18 | 73.87% | **9.03%** | 212.53% |
| NGM (Wang et al., 2021a) | 15.72 | 62.35% | 17.78% | 129.75% |
| + LR | 69.57 | 54.41% | 16.54% | 117.36% |
| RGM w/o rev. | 47.24 | 41.49% | 12.58% | 119.75% |
| + LR | 96.45 | 39.70% | 11.65% | 109.47% |
| RGM | 75.53 | **35.84%** | 10.74% | **101.13%** |

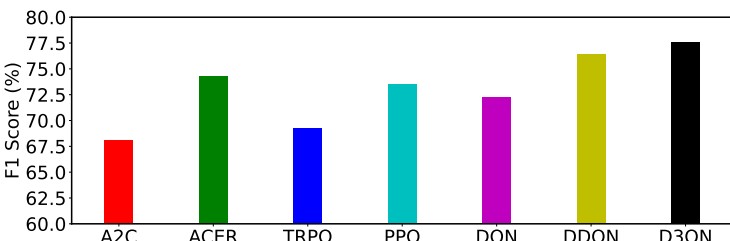

Figure 11: Average F1 score (↑) of different RL algorithms, on five classes of Willow Object with 10 inliers and 3 outliers in both sides for matching.

Table 8: Quantile of objective score and F1 score of the solutions found by RGM and RRWM on bus category in Pascal VOC without outlier. The discrepancy of F1 score and objective score is clear.

| | Objective range | (0.00, 0.99) | [0.99, 1.00) | = 1.00 | (1.00, 1.50) |
|---|---|---|---|---|---|
| RRWM | Proportion | 38.4% | 20.0% | 11.9% | 29.7% |
| | F1 score | 42.1% | 68.6% | 100.0% | 62.4% |
| RGM | Proportion | 4.0% | 11.6% | 45.0% | 39.4% |
| | F1 score | 40.6% | 61.2% | 100.0% | 56.5% |

improve the performance, which can be useful when the matching can be conducted by manual annotation in the beginning.

### A.4.6 ABLATION STUDY: REVOCABLE ACTION V.S. LOCAL REWRITE

We study the effectiveness of our revocable action scheme, by comparing it with the local rewrite (LR (Chen & Tian, 2019)) under the same RL framework. LR is an influential mechanism in RL-based combinatorial optimization. It tries to improve a given solution instead of generating one from scratch. To some extent, LR can also reverse the applied actions by its local rewrite mechanism and it is recognized as state-of-the-art technique for improving RL.

We conduct comparative experiments on QAPLIB. We use LR to improve the solution given by the baselines and RGM without the revocable scheme (RGM w/o rev.), and compare the results with RGM. Table 7 shows that our revocable framework RGM still performs the best compared to all boosted baselines. It turns out LR does improve the original solutions, but its performance and efficiency is still below our revocable framework.

### A.4.7 ABLATION STUDY: USING ALTERNATIVE RL BACKBONES

In RGM, we adopt Double Dueling DQN (D3QN) with priority experience replay as our backbone, for which we perform ablation study against alternatives on Willow Object with 10 inliers and 3 outliers. Fig. 11 shows the mean F1 score over five classes. We compare D3QN with popular backbones: A2C (Mnih et al., 2016), ACER (Wang et al., 2017), TRPO (Schulman et al., 2015), PPO (Schulman et al., 2017), the original DQN (Mnih et al., 2013), and Double DQN (Hasselt et al., 2016). D3QN outperforms all other algorithms in the Willow Object. We think that the main reason is that: graph matching is a discrete decision making problem, where the value based methods (DQN, DDQN, D3QN) can be more suitable than the policy based methods, which is widely accepted (Sutton & Barto, 2005; Ivanov & D'yakonov, 2019; Sutton & Barto, 2018).

### A.4.8 INCONSISTENCY BETWEEN AFFINITY OBJECTIVE AND F1

Finally, we discuss a standing issue in graph matching and possibly also in other optimization tasks regardless the presence of outliers. The matching accuracy or F1 score is inconsistent with the value of objective function. In our analysis, the reason is probably that the objective function cannot perfectly model the ultimate goal, due to limited modeling capacity and noise etc. For example, as shown in Table 8, when applying our RGM and RRWM (or any other method is fine from the different-quality solution sampling perspective) on the real image dataset Pascal VOC, the resulting quantile statistics about the objective score deliver an important message: $39.4\%$ sampled solutions by RGM can achieve an objective score even higher than one, which means these wrong solutions can even get a higher score than the ground truth matching. Note that the front-end features CNN/GNN and affinity metric model are learned by the state-of-the-art supervised model BBGM, however the learned affinity function seems still not perfectly fit with the F1 score. In fact, this mismatch issue relates to the front-end affinity learning and outlier dismissing, which is not the scope of the back-end solver as focused in this paper. As Table 8 shows, though there are $39.4\%$ instances whose affinity scores are larger than 1, at the other $60.6\%$ instances, our RGM can solve three-quarters of these instances perfectly. Compared to RRWM, our RGM can solve more instances with the affinity score equals to or nears to one, and that's why our RGM can outperform RRWM and other baselines. Despite its impressive results achieved in our paper, it also suggests the limitation of RL-based solvers which pursuits the high objective score which can sometimes be biased. One possibly mitigation way is involving multiple graphs (Yan et al., 2016; Jiang et al., 2021) which we leave for future work.

