# OpenReview forum: "Revocable Deep Reinforcement Learning with Affinity Regularization for Outlier-Robust Graph Matching"
_ICLR.cc/2023/Conference — ICLR 2023 poster_

### Official Review · Reviewer_6KL8 · 2022-10-24

**Confidence:** 4
**Correctness:** 3
**Technical Novelty And Significance:** 3
**Empirical Novelty And Significance:** 3
**Recommendation:** 8

**Clarity, Quality, Novelty And Reproducibility:**

In general, the elaboration and formulation is clear so that a broad range of readers can grasp the general idea of the paper. However, several typos are expected to be revised and some detailed formulae need to be better enunciated. This is a novel work that first utilizes reinforcement learning in the back-end solver of graph matching. By regarding GM as a sequential selection task, the model shows robustness for outliers even in the absence of label. Details in the appendix provides a promising reproducibility for this work.

**Strength And Weaknesses:**

Strength:
1. Training of Reinforcement-based method proposed in this paper does not require labelled data which is costly and time-consuming in large-scale datasets.
2. The proposed method is a back-end solver for the most general form of GM with Lawler’s QAP as the input, which means it’s a plug-and-play learning-based module for multiple front-end models.
3. Extensive experiments are conducted to proffer a comprehensive analysis.

Weakness:
1. In “chr (12-25)” and “lipa (20-60)” column of Table 4, why the model SK-JA overperforms your proposed method so much?
2. Can you provide some experimental results when the size of training set varies?
3. As for formulae, Eq (7) has a wrong sign. And there exist several pieces that I am unable to understand. “F,W ∈ R^{n1×n2}” contradicts with Eq.2 above which indicates F and W are matrices of n1*n2 rows and n1*n2 columns instead of a matrix with n1 rows and n2 columns. In Eq. 9, h1 = X’ • θ1^T where X’ is a permutation matrix which is a 0-1 matrix with a shape of (n1, n2). But h1 is of the shape (n1*n2, ). So I wonder how it can be made. Still in Eq. 9, on the RHS of the calculation of h4, W is an edge weight matrix which is ought to be a matrix in R^{n1*n2 × n1*n2} according to the corrected  formula “F,W ∈ R^{n1×n2}”. However, “θ5 ∈ R^d” as is written in the paper. I cannot understand how these two matrices multiplied.
4. Several grammar mistakes and typos are made in the paper, e.g., “learnable features are shown more expressive” (probably means “have shown”?), “a also”, “the our”, “treat is as a constant”, “Wwe”, etc.

**Summary Of The Paper:**

This paper introduced deep reinforcement learning into the study of graph matching to tackle the ubiquitous outliers in practical application. Traditional back-end solvers generate the matching result between graphs in a single forward propagation attempting to maximize affinity score. However, the blindly pursuing higher affinity score induces to involve outliers because all elements in the affinity matrix are positive in most cases. Therefore, graph matching is modelled as a sequential selection task on the association graph in this paper, where the agent is expected to early stop when inliers are all selected instead of selecting the left outliers. In order to make their methods robust and prevent the agent from selecting superfluous vertices, the authors propose two strategies:
1. to hardcoding an exact number of inliers at which the agent stops as soon as it achieves,
2. to updating a regularized affinity matrix where the score of overly selection is discounted.
Additionally, a simple yet effective revokable action mechanism is put forward to give the agent the opportunity to “regret” its past suboptimal decision.
Finally, with all the aforementioned techniques, RGM shows competitive result on both image data and pure combinatorial optimization instances.

**Summary Of The Review:**

This paper proposes a deep RL-based model for robust and unlabeled graph matching. Revokable action mechanism and affinity regularization is proposed to fit with the RL schema and further boost the robustness. Exhaustive experiments have been conducted to prove the effectiveness of their method. Additionally, a fact is observed that there does not exist a constant positive correlation between our target metric, F1 score, and the optimized metric, objective score, which has a potential inspiration for other researches for further study.

---

> ### Author Response · Authors · 2022-11-11
> **Response to Reviewer 6KL8**
>
> We appreciate the reviewer for the detailed comments on our paper. Thank you for your recognition of our work. We set out below our responses to each of the questions you may concern about.
>
> > ***Q1: "In “chr (12-25)” and “lipa (20-60)” column of Table 4, why the model SK-JA overperforms your proposed method so much?"***
>
> > In the experiments on QAPLIB, we find that the solutions found by RGM are sometimes different from the solutions found by traditional solvers (SK-JA). For example, in instance "Esc.b" and "Ecs.h", our RGM can find the optimal solutions with the same minimal objective score as provided on the QAPLIB website, however, the solutions found by RGM is totally different from the provided solutions. Therefore, we think the solving characteristics of RGM and traditional solvers may be different. Therefore, it is acceptable that RGM does not outperform SK-JA on "chr (12-25)" and "lipa (20-60)", since these two categories may be more suitable for traditional solvers. When considering all the categories in QAPLIB, our RGM has a clear advantage over the baselines.
>
> > ***Q2: "Can you provide some experimental results when the size of training set varies?"***
>
> > Please note that the size of the training set does vary in our experiments on QAPLIB. As mentioned in the last paragraph, the caption "chr(12-25)" denotes the graph size varies from 12 to 25 in the training set of the category "chr". Besides, the graph size in Pascal VOC is also not fixed in each category, for example, the "car" category contains images with 5 to 8 keypoints.
>
> > ***Q3: Equation mistake in Eq.9***
>
> > Sorry for the confusion, we check our equation again and find the definition of $\mathbf{F}$ and $\mathbf{W}$ should be corrected to $R^{n1n2 \times n1n2}$ instead of $R^{n1 \times n2}$, as you suggested in the comments. Thanks for pointing out our mistake. As for the calculation of $h_{4}$, we miss a transpose on $\theta_{5}$ (sorry again), and the dimension operation should be: $\sum(n1n2 \times n1n2 \times 1) * (d \times 1)^{\top} = \sum(n1n2 \times n1n2 \times d) = (n1n2 \times d)$. Here the sum operation removes the second dimension.
>
> > ***Q3: Grammar mistake***
>
> > Sorry again for the inconvenience. We will modify the paper and double-check the whole paper again. Thanks for your kind reminder.

---

### Official Review · Reviewer_bTMW · 2022-10-24

**Confidence:** 4
**Correctness:** 3
**Technical Novelty And Significance:** 3
**Empirical Novelty And Significance:** 3
**Recommendation:** 6

**Clarity, Quality, Novelty And Reproducibility:**

[Clarity]: Neutral
[Quality]: Good
[Novelty ]: Good
[Reproducibility]: Fair

**Strength And Weaknesses:**

[Strength]

1. To the best of my knowledge, using deep reinforcement learning for graph matching is novel;
2. Experimental results are good.

[Weaknesses]

I cannot find major weaknesses, and only have minor comments.

1. The proposed method can only handle very small-scale graph matching problem (almost toy examples). It is very hard to extend the proposed method for large-scale graph matching (~2000 nodes for each graph).

The reason is that the number of vertices in the association graph would be 400 millions.

Probably I should not blame authors for this limitation since there are some previous works on PASCAL VOC.

2. It is partially true to say majority works (Zanfir et al., 2018; Wang et al., 2021a) that assume there is at most one graph
containing outliers.  For example, using the sinkhorn solver with padded slack row and column would naturally solve matching when outlier s exist in both two graphs.

I would expect authors to compare the proposed method with sinkhorn solver (padding slack row and column) in [Learning Feature Matching with Graph Neural Networks].

3. If we remove the second order (edge) affinities, what is the performance of the proposed method?

4. Please provide time comparisons of the proposed method with respect to state-of-the-art methods.


**Summary Of The Paper:**

This paper introduces a sequential node matching scheme for graph matching, via deep reinforcement learning.

The proposed scheme differs from the majority of existing works that obtain the whole matching in one shot.

The main effectiveness of the proposed method seems to lie in handling outliers.

Experiments on both synthetic and real-world datasets show the effectiveness of the proposed method.



**Summary Of The Review:**

Though it is novel to introduce deep reinforcement learning, the proposed method can only work for toy-scale graph matching and the comparison with respect to sinkhorn solver (padding slack row and column) is missing.

---

> ### Author Response · Authors · 2022-11-11
> **Response to Reviewer bTMW**
>
>  ***Table 1: Performance Comparison on Willow Object Dataset with 1-6 Outliers w.r.t F1 score***
>
> |                   | 1      | 2      | 3      | 4      | 5      | 6      |
> | ----------------- | ------ | ------ | ------ | ------ | ------ | ------ |
> | BBGM              | 87.21% | 79.84% | 73.41% | 69.09% | 61.79% | 58.07% |
> | NGM-v2            | 86.84% | 78.90% | 73.23% | 69.96% | 60.52% | 56.73% |
> | NGM-v2-plus       | 87.03% | 79.33% | 74.07% | 70.52% | 62.86% | 60.57% |
> | NGM-v2-plus-x5    | 87.19% | 79.73% | 74.41% | 70.75% | 63.10% | 60.95% |
> | RGM               | 87.68% | 80.37% | 77.57% | 73.77% | 67.49% | 65.10% |
> | RGM w/o 2nd order | 84.02% | 79.46% | 73.17% | 71.14% | 63.97% | 62.51% |
>
> Thank you for your valuable comments which offer us a lot to explore and we deeply cherish. Here, we try our best to answer the questions from your comments and hope we can alleviate your concerns.
>
> > ***Q1: The scalability issue***
>
> > - Admittedly, scalability is indeed a significant issue. While in our humble opinion, we think that the scalability issue actually lies in the whole graph matching field. In fact, our RGM does not raise additional burden, since all deep GM methods that utilize the association graph or Lawerl's QAP formulation will suffer the same $O(n^4)$ cost, including the SOTA BBGM or NGM-v2.
> > - To the best of our knowledge, the max graph size of the commonly used datasets in existing GM methods is up to 30, including Willow Object, Pascal VOC and etc. In our paper, the experiments scale of RGM is up to 64, and we think it is quite a large scale in the graph matching area.
> > - We admit that the sequential decision making process of RGM does require more time compared to the one-shot learning deep GM methods, but the memory cost is the same, which means the usability and space complexity of RGM are the same as the existing deep GM methods (NGM-v2, BBGM). As the experiments show the performance of RGM, we consider RGM is the most scalable solution for the neural solver in the outlier GM for now.
>
> > ***Q2: Improve the Sinkhorn via the approach in SuperGlue***
>
> > Thanks for the inspiration of using the idea of SuperGlue by padding the slack row and column to the Sinkhorn. We modify the code of NGM-v2 accordingly and name it "NGM-v2-plus", and we test the performance of "NGM-v2-plus". As Table 1 shows, the idea of SuperGlue does improve the performance of the original NGM, but the gap to our RGM still remains. We are surprised by the effectiveness of such modification, and we think we can try to utilize this mechanism in our own framework in the future.
>
> > ***Q3: "If we remove the second order (edge) affinities, what is the performance of the proposed method?"***
>
> > As for showing the effectiveness of the second order affinities, we modify our RGM and remove the second order affinities in the affinity matrix input (namely "RGM w/o 2nd order"). We compare the "RGM w/o 2nd order" with the origin RGM and other baselines in Table 1. We can see that without the usage of second order affinities the performance of RGM will drop up to 4%, which shows the necessity of the second order affinity to some extent.
>
> > ***Q4: "Please provide time comparisons of the proposed method with respect to state-of-the-art methods"***
>
> > We have a time comparison of RGM and the baselines in Table 4 in the submitted paper, where the time cost of our RMG is about 4 to 5 times of the SOTA NGM. We think this extra time cost is acceptable considering the performance advantage of RGM. Besides, we conduct a new experiment that makes the "NGM-v2-plus" run 5 times and maintain the highest score, namely "NGM-v2-plus-x5". We can see that our RGM can still outperform "NGM-v2-plus-x5" while the time cost gap between RGM and "NGM-v2-plus-x5" is relatively small.

---

### Official Review · Reviewer_QGQe · 2022-11-04

**Confidence:** 5
**Correctness:** 4
**Technical Novelty And Significance:** 2
**Empirical Novelty And Significance:** 3
**Recommendation:** 5

**Clarity, Quality, Novelty And Reproducibility:**

Clarity:
I found the paper to be quite clear and easy to follow.

Quality:
In my opinion, the proposed approach is sound and the experiment results are reasonable and supports the proposed approach.

Novelty:
As discussed in “strengths”, I think the “revocable” mechanism and the approximation of non-QAP affinity score into a QAP formulation are interesting and novel.  The RL formulation and the regularized QAP formulation are more standard.

Reproducibility:
The paper contains a reasonable level of details with an extensive appendix, which are good for reproducibility.  On the other hand the system contains a fair amount of complexity, which makes it harder to reproduce and apply in general.

**Strength And Weaknesses:**

Strengths:
* The paper is quite clear and easy to follow.
* The “revocable” mechanism seems new.
* The approximation of the non-QAP affinity score into a QAP formulation is interesting and could be a convenient and fairly general way to handle other types of optimization problems.
* Experiment results show that the proposed approach reaches higher matching quality overall, compared to prior approaches.

Weaknesses:
* The most significant weakness is that the proposed approach and QAP formulation is very expensive, $O(n_1^2 n_2^2)$, which is $O(n^4)$ for comparable $n_1$ and $n_2$, for each step, and the proposed approach is fundamentally limited to do graph matching for very small graphs.  The largest graph considered in the experiment contains up to just 64 nodes, and I don’t see a clear way to scale this up to instances of significantly larger scale.  Therefore the applicability of this approach and formulation seems to be very limited.
* The proposed solution contains a collection of smaller bits and the entire system has a fair amount of complexity to build.
* Structure of the paper could be improved - introduction seems to be a bit too long, while many results have to be left in the appendix.

**Summary Of The Paper:**

This paper presents an approach for solving the quadratic assignment (QAP) graph matching problem, which matches nodes across two graphs and the matching quality is measured with the affinity of both the matched nodes as well as the matched edges.  The approach is a collection of smaller things, including (1) an RL formulation of the sequential decision process that matches one pair of nodes at each step; (2) a “revocable” mechanism that allows each step to overwrite previously matched pairs; (3) a regularized affinity score to handle outliers and (4) an approximation scheme to convert the regularized affinity score back into a QAP formulation.

Experiments on synthetic tasks, key-point matching tasks in the computer vision domain and on the standard QAP instances from QAPLIB show that the proposed approach seems to be working well.

**Summary Of The Review:**

I have reviewed this paper twice before for another venue, and it has improved a lot since then, with many of the reviewers’ suggestions incorporated in this submission.

This paper as it stands now has clear strengths and weaknesses, with a few novel ideas and experiments showing that this approach does outperform previous work quite consistently; on the other hand, scalability seems like a pressing issue that significantly affects the applicability of this approach.  Given the state of the current field of computer vision where key point matching can already handle orders of magnitude more keypoints than this proposed approach is capable of doing, I think this paper won’t have an immediate impact on the field.  But once the scalability problem is alleviated, or with a better matched application where the QAP formulation is really required, this approach could have a larger impact.

Overall I’m neutral about the paper and wouldn’t mind either accepting or rejecting it.

---

> ### Author Response · Authors · 2022-11-11
> **Response to Reviewer QGQe**
>
>  ***Table 1: Performance Comparison on Willow Object Dataset with 1-6 Outliers w.r.t F1 score***
>
> | |1|2|3|4|5|6|
> |-|-|-|-|-|-|-|
> | BBGM | 87.21% | 79.84% | 73.41% | 69.09% | 61.79% | 58.07% |
> | NGM-v2| 86.84% | 78.90% | 73.23% | 69.96% | 60.52% | 56.73% |
> | NGM-v2-plus| 87.03% | 79.33% | 74.07% | 70.52% | 62.86% | 60.57% |
> | NGM-v2-plus-x5| 87.19% | 79.73% | 74.41% | 70.75% | 63.10% | 60.95% |
> | RGM  | 87.68% | 80.37% | 77.57% | 73.77% | 67.49% | 65.10% |
>
> Thank you for your review and for kindly remembering our paper! We really appreciate your acknowledgment of our work, especially our improvement since the last venue. Here, we answer the questions from your comments and try our best to alleviate your concerns. We remain available for your further queries if needed.
> > ***Q1: The scalability issue***
>
> > - As you mentioned in the comments, scalability is indeed a significant issue. However, in our humble opinion, we think that the scalability issue actually lies in the whole graph matching field which is NP-hard. In fact, our proposed RGM does not raise additional burden, since all deep GM methods that utilize the association graph or Lawerl's QAP formulation will suffer the same $O(n^{4})$ cost, including the SOTA BBGM or NGM-v2.
> > - To the best of our knowledge, the max graph size of the commonly used datasets in existing GM methods is up to 30, including Willow Object, Pascal VOC and etc. In our paper, the experiments scale of RGM is up to 64, and we think it is quite a large scale in the graph matching area. We admit that the sequential decision making process of RGM does require more time compared to the existing one-shot deep GM methods, but we share the same memory cost, which means the usability and space complexity of RGM are the same as the existing deep GM methods (NGM-v2, BBGM).
> > - Moreover, GM with outliers is more difficult than the original GM itself, and we mainly target this scenario with additional optimizations for two-side outlier scenario which is even harder than the NP-hard vanila GM. Even compared with the outlier version or the multiple-run version of NGM-v2 (the variant of RGM combined with Superglue suggested by Reviewer bTMW), our method still reaches higher F1 scores. As Table 1 shows, the "NGM-v2-plus-x5" denotes running the outlier version of NGM-v2 for 5 times with random initialization and maintaining the highest score, which has a similar time cost as RGM but our RGM still performs better. Therefore, we consider RGM is the most scalable solution for the neural solver in the outlier GM for now.
>
> > ***Q2: "The proposed solution contains a collection of smaller bits and the entire system has a fair amount of complexity to build"***
>
> >Admittedly, we did propose several (complex) components in RGM which cause a lot of space to describe in our paper, sorry for the inconvenience. Since the outlier GM is much more difficult than the original GM, we then propose these components to solve the outlier GM step by step, including the revocable mechanism, the inlier count information, the affinity regularization and etc. Though there are multiple components in RGM, they are integrated into one single framework and we can simply run one single file to train/test RGM. As a result, the running of RGM is not hard to complete to some extent.
>
> > ***Q3: "Structure of the paper could be improved"***
>
> > Sorry for the crowded structure of our paper. As mentioned in the last paragraph, there are multiple key components in RGM and we have to spend a lot of space introducing them, plus we conduct various experiments to demonstrate the performance of RGM (some of the experiments are suggested by the reviews in the venue before, thanks for the advice), so we have to move some content into the appendix. We are sorry for the inconvenience.
>
>
> **Highlight**
>
> Finally, we would like to highlight the necessity of our work. Please note that the GM problem has been intensively studied for at least 50 years[1], and little work is really devoted to this important problem: graph matching with outliers using machine learning (deep reinforcement learning). For such a challenging NP-hard problem, which is even much harder than standard NP-hard graph matching without outliers, we think it calls for continuous efforts in a long journey to push forward the frontier of this area. As it is still an open problem for achieving both accuracy (F1 score) and scalability at this point, we hope our efforts can be a starting point to attract more people in the community to work on this problem, and probably by readily using our released protocols, preprocessed datasets, and source code.
>
> [1] Emmert-Streib F, Dehmer M, Shi Y. Fifty years of graph matching, network alignment and network comparison[J]. Information sciences, 2016, 346: 180-197.

---

> ### Author Response · Authors · 2022-12-04
> **Response to Reviewer QGQe (12.4 Update)**
>
> Thanks for your (multi-round) time and efforts in reviewing our paper which indeed helps us improve the paper a lot. In our response, we tried to address your questions including additional experimental results. We are looking forward to your further comments which are sincerely appreciated.
>
> For the scalability concern, we think it is always a challenge for combinatorial and graph algorithms which often encounter the NP-hard issue, where graph matching or QAP is not an exception. On the other hand, we have shown that our method RGM can successfully handle a scale larger than Pascal VOC, which is the biggest-scaled benchmark in most existing GM literature.
>
> Specifically, our RGM is 3-5 times slower than the SOTA NGMv2, and we have found in many combinatorial optimization papers' experiment settings, such a time overhead gap is common for compared methods.
>
> The advantage of our method is its flexibility to handle the arbitrary number of outliers in both graphs, which is a **very** practical case, yet cannot be achieved in the original paper of NGMv2, nor by existing learning-based GM methods. We have shown in our added experiments that, even running 5 times of NGMv2, RGM still outperforms.
>
> Another flexibility of our method is the any-time algorithm character that it can always provide so-far obtained node matchings (a subset of the complete solution) which is an attractive advantage as long shown in related literature [1-4]. In contrast, existing GM neural models always require users to wait until it outputs a complete solution. They are not any-time algorithms. We believe the anytime advantage is a useful feature in face of large-scale problems in the sense that one can always require and fetch an intermediate result no matter how large-scale the problem is. This can be often leveraged in downstream applications see references [5-7]. We plan to add more content about the any-time algorithm in the camera-ready vision of our paper. (since the pdf update period is passed)
>
> In a nutshell, RGM is an effort to make practical graph matching in the wild against outliers. While existing GM works are a bit in a toy setting by assuming at most only one side of the graph contains unmatchable outliers (or without outliers).
>
> [1] Zilberstein S. Using anytime algorithms in intelligent systems[J]. AI magazine, 1996, 17(3): 73-73.
>
> [2] Likhachev M, Ferguson D, Gordon G, et al. Anytime search in dynamic graphs[J]. Artificial Intelligence, 2008, 172(14): 1613-1643.
>
> [3] Vlassis N, Elhorst R, Kok J R. Anytime algorithms for multiagent decision making using coordination graphs[C]//2004 IEEE International Conference on Systems, Man and Cybernetics (IEEE Cat. No. 04CH37583). IEEE, 2004, 1: 953-957.
>
> [4] Likhachev M, Ferguson D I, Gordon G J, et al. Anytime Dynamic A*: An Anytime, Replanning Algorithm[C]//ICAPS. 2005, 5: 262-271.
>
> [5] Ueno K, Xi X, Keogh E, et al. Anytime classification using the nearest neighbor algorithm with applications to stream mining[C]//Sixth International Conference on Data Mining (ICDM'06). IEEE, 2006: 623-632.
>
> [6] Kranen P, Seidl T. Harnessing the strengths of anytime algorithms for constant data streams[J]. Data Mining and Knowledge Discovery, 2009, 19(2): 245-260.
>
> [7] Hong S, Lee S U, Huang X, et al. An anytime algorithm for chance constrained stochastic shortest path problems and its application to aircraft routing[C]//2021 IEEE International Conference on Robotics and Automation (ICRA). IEEE, 2021: 475-481.

---

### Author Response · Authors · 2022-11-18
**Looking forward to your feedback**

Dear reviewers, given the DDL of the discussion stage 1 is approaching,  we are sincerely looking forward to your reply and we could provide more responses if needed.

---

### Decision · Program_Chairs · 2023-01-20

**Decision:**

Accept: poster

**Justification For Why Not Higher Score:**

Scalability is a major issue that prevents the results of this paper from being widely applicable and having a large impact.

**Justification For Why Not Lower Score:**

Some ideas of the paper is new and experiment results are good.

Reviewers overall favor acceptance.

**Metareview: Summary, Strengths And Weaknesses:**

This paper presents a sequential decision process for solving the quadratic assignment problem (QAP).  The model is learned with RL, and a “revocable” mechanism is introduced to allow each step of the process to overwrite the decisions made previously.

Overall reviewers are on the accept side, and acknowledge that the proposed RL for QAP approach is novel, and the experiment results are good.

A couple reviewers raised the scalability issue, which is the biggest issue of this paper and the QAP formulation in general.  The authors should probably think more about the practical usability of this approach, and whether the trade-off between a more expressive but much more expensive QAP formulation and a less expressive but way more efficient matching formulation is worthwhile, and if so in which applications this is really worth doing.

**Note From Pc:**

if the above contains the word "oral" or "spotlight" please see: "oral" presentation means -> notable-top-5% and "spotlight" means -> notable-top-25%. As stated in our emails, we are disassociating presentation type from AC recommendations